# Activating lattice oxygen in high-entropy LDH for robust and durable water oxidation

Fangqing Wang[1,2], Peichao Zou[3], Yangyang Zhang [2], Wenli Pan[4], Ying Li[1,2], Limin Liang[2], Cong Chen[2], Hui Liu [1,2] ✉ & Shijian Zheng[1,2] ✉

The oxygen evolution reaction is known to be a kinetic bottleneck for water splitting. Triggering the lattice oxygen oxidation mechanism (LOM) can break the theoretical limit of the conventional adsorbate evolution mechanism and enhance the oxygen evolution reaction kinetics, yet the unsatisfied stability remains a grand challenge. Here, we report a high-entropy MnFeCoNiCu layered double hydroxide decorated with Au single atoms and O vacancies ($Au_{SA}$-MnFeCoNiCu LDH), which not only displays a low overpotential of 213 mV at 10 mA cm$^{-2}$ and high mass activity of 732.925 A g$^{-1}$ at 250 mV overpotential in 1.0 M KOH, but also delivers good stability with 700 h of continuous operation at ~100 mA cm$^{-2}$. Combining the advanced spectroscopic techniques and density functional theory calculations, it is demonstrated that the synergistic interaction between the incorporated Au single atoms and O vacancies leads to an upshift in the O 2$p$ band and weakens the metal-O bond, thus triggering the LOM, reducing the energy barrier, and boosting the intrinsic activity.

Hydrogen generation through water electrolysis is an ideal way to utilize and store intermittent renewable energy sources, such as solar and wind power, which can effectively tackle the energy crisis and carbon emission issues[1,2]. However, the anodic oxygen evolution reaction (OER) involved in the process presents a complex four-electron transfer process coupled with protons transfer, leading to sluggish kinetics that limit the water splitting process[3,4]. Hence, a fundamental understanding of the OER mechanism is essential for boosting OER kinetics and exploring highly efficient and durable OER electrocatalysts. According to the conventional adsorbate evolution mechanism (AEM), the adsorption energies of the OER intermediates (*OH and *OOH) on metal active sites follow a linear relationship ($\Delta G_{OOH} = \Delta G_{OH} + 3.2 \pm 0.2$ eV), which sets a theoretical limit on the overpotential of around 370 mV[5–7]. Recently, a lattice oxygen oxidation mechanism (LOM) has been proposed, which involves the activation and redox of the lattice oxygen during water oxidation, and can circumvent the limitation of the linear AEM relationship, thereby

reducing the energy barrier[8,9]. Therefore, it's highly desirable to exploit robust and durable OER electrocatalysts based on LOM, which has become a research hotspot in the field of water splitting[10,11].

Transition-metal oxides/(oxy)hydroxides have been extensively studied for their potential to trigger the LOM during OER. These include defect-rich $RuO_2$, (La, Sr)$CoO_3$, CoZn (oxy)hydroxides, $CoAl_2O_4$, $NaxMn_3O_7$, NiFe (oxy)hydroxides, MoNiFe (oxy)hydroxides et al. [7,12–14]. Experimental and theoretical calculation studies have shown that upshifting the O 2$p$ band to approach the Fermi level ($E_F$) is the key to triggering the LOM of transition-metal oxides/hydroxides. This leads to increased orbital overlap between the O 2$p$ band and metal (M) $d$ band, and strengthens the M-O covalent bond, which makes the redox of lattice oxygen more thermodynamically favorable[15,16]. However, the participation of the lattice oxygen in OER electrocatalysts can result in the leaching of the metal species on the surface that induces the collapse or phase transition of the bulk phase, leading to poor stability[11]. As a result, obtaining high catalytic activity

[1]Key Laboratory of Special Functional Materials for Ecological Environment and Information (Ministry of Education), Hebei University of Technology, Tianjin 300130, PR China. [2]School of Material Science and Engineering, Hebei University of Technology, Tianjin 300130, PR China. [3]Department of Physics and Astronomy, University of California, Irvine, CA 92697, USA. [4]Graduate School of Human and Environmental Studies, Kyoto University, Yoshida-nihonmatsu-cho, Sakyo, Kyoto 606-8501, Japan. ✉e-mail: liuhui2013@hebut.edu.cn; sjzheng@hebut.edu.cn

and good stability simultaneously, especially under large-current-density OER conditions, remains a grand challenge for most OER electrocatalysts based on LOM[17].

High-entropy materials (HEMs), such as high-alloy oxides, LDH, sulfides, and fluorides, are now emerging as a versatile platform for electrochemical OER due to their unique properties, including the high-entropy effect, cocktail effect, and sluggish diffusion effect[18–23]. Compared to traditional unitary, binary, and ternary OER electrocatalysts, HEMs exhibit similar catalytic activity but better stability[24], which may be attributed to the entropy-stabilized effect and sluggish diffusion effect that prevent phase transition and metal leaching[25]. Motivated by these findings, we hypothesize that triggering lattice oxygen activation in HEMs electrocatalysts may enable to obtain both high activity and good stability that traditional LOM-based OER electrocatalysts can't do. Previous studies validate that increasing the covalency of metal-oxygen bonds is critical to triggering lattice-oxygen oxidation[26]. Given that the covalency of metal-oxygen bonds is determined by the electronegativity of metal, and Au has the greatest electronegativity (2.54) of all metals, we speculate the incorporation of Au atoms into high-entropy LDH may increase metal-oxygen covalency, triggering LOM. Also, Au is a kind of inert metal element that has strong alkali resistance and electrochemical stability[27], possibly avoiding the leaching during anodic oxidation, which is beneficial for the structure stability of the catalyst. Additionally, the introduction of a small amount of Au single atoms on the surface of high-entropy LDH instead of doping in the lattice with large amounts of Au atoms may be more realistic in consideration of the high cost of Au and the big difference in ion radius between Au and 3d transition elements[28].

In this study, we report a high-entropy electrocatalyst, MnFeCoNiCu LDH decorated with Au single atoms and oxygen vacancies (denoted as $Au_{SA}$-MnFeCoNiCu LDH), which exhibits both high OER activity and promising stability. The hard X-ray absorption fine structure spectroscopy (XAFS) and high-angle annular dark field-scanning transmission electron microscopy (HAADF-STEM) characterizations demonstrate that the atomically dispersed Au atoms are doped in Fe sites, while X-ray photoelectron spectroscopy (XPS) and electron paramagnetic resonance (EPR) measurements validate the existence of oxygen vacancies. The combination of in situ Raman spectroscopy, $^{18}O$ isotope labeling mass spectroscopy, and density functional theory (DFT) calculations suggest that the synergistic effect of doped Au single atoms and generated oxygen vacancies transforms the OER mechanism of MnFeCoNiCu LDH from AEM to LOM, which reduces the energy barrier and enhances the intrinsic activity. The $Au_{SA}$-MnFeCoNiCu LDH catalyst exhibits a significantly enhanced OER activity with an overpotential of 213 mV at 10 mA cm$^{-2}$, which is 110 mV lower than that of pristine MnFeCoNiCu LDH. Most importantly, $Au_{SA}$-MnFeCoNiCu LDH shows remarkable stability, sustaining 700 h at ~100 mA cm$^{-2}$ with only 6.4% degradation. This work not only successfully regulate the OER pathway of high-entropy LDH from traditional AEM to LOM, but also sheds light on the active mechanism on the lattice oxygen in high-entropy LDH, providing a way for the design of high-entropy-based OER electrocatalysts with high activity and durability based on LOM.

## Results
### Synthesis and structural characterizations
First, the high-entropy MnFeCoNiCu LDH electrocatalyst was prepared using a typical hydrothermal method (see schematic in Fig. 1a). X-ray diffraction (XRD) analysis shows that the MnFeCoNiCu LDH conforms to the hexagonal hydrotalcite structure (PDF#50-0235) (Fig. S1a). In addition, the MnFeCoNiCu LDH exhibited a nanosheet morphology with smooth surfaces and sharp edges, as observed in the field emission scanning electron microscope (FE-SEM) and transmission electron microscope (TEM) images (Fig. S1b, c). The energy dispersion spectrum (EDS) result and element mapping images (Fig. S1d, e) show

that Mn, Fe, Co, Ni, Cu, O elements were uniformly distributed throughout a typical nanosheet, with the molar content of five metal elements (Mn, Fe, Co, Ni and Cu) being 3.31 %, 7.88%, 4.16 %, 8.54 % and 7.38 %, respectively (Table S1, calculated from the EDS). All these results indicate that we have successfully prepared a high-entropy MnFeCoNiCu LDH nanosheet electrocatalyst through a facile hydrothermal method.

Subsequently, Au single atoms were incorporated into MnFeCoNiCu LDH by an electrochemical cyclic voltammetry (CV) method. XRD analysis shows that the introduction of Au single atoms did not change the crystalline structure of MnFeCoNiCu LDH (Fig. S2). FE-SEM and TEM images (Fig. S3a and Fig. 1b) validate that $Au_{SA}$-MnFeCoNiCu LDH still maintained the nanosheet structure, and no obvious metal or metal oxide particles were observed, implying that Au atoms may exist in the form of single atoms. Atomic force microscopy (AFM) results show that a typical $Au_{SA}$-MnFeCoNiCu LDH nanosheet has a thickness of 3.4 nm (Fig. 1c). Recent studies have demonstrated that the thickness of single layer of high-entropy LDH is about 0.7–0.9 nm[29–31]. Moreover, the layer spacing of LDH can be reflected by the diffraction peak of (003) plane in the XRD pattern (Fig. S2)[32,33], and the calculated value based on Bragg formula is 0.74 nm. Therefore, the as-prepared $Au_{SA}$-MnFeCoNiCu LDH is comprised of 3 layers. The clear lattice fringe with an interplanar distance of 0.261 nm shown in the high-resolution TEM (Fig. 1d) corresponds to the (101) plane of $Au_{SA}$-MnFeCoNiCu LDH. In contrast, the lattice spacing of MnFeCoNiCu LDH is 0.258 nm (Fig. S4). The slight expansion of $Au_{SA}$-MnFeCoNiCu LDH in terms of interplanar distance relative to MnFeCoNiCu LDH may be attributed to the larger atomic radius of Au than other metals. Moreover, the selected area electron diffraction pattern (Fig. 1e), collected along the [12$\bar{1}$] crystal belt axis, also demonstrates the single crystal property of $Au_{SA}$-MnFeCoNiCu LDH with a hexagonal hydrotalcite structure (PDF#50-0235), agreeing well with the XRD result. Aberration-corrected high-angle annular dark-field scanning transmission electron microscope (AC-HAADF-STEM) images (Fig. 1f) confirm the atomic-level dispersion of Au atoms on the MnFeCoNiCu LDH substrate. Notably, the bright Au atoms (marked with yellow circles) were regularly located at the lattice point of metal atoms in MnFeCoNiCu LDH, suggesting that Au atoms may have replaced the metal atoms in MnFeCoNiCu LDH or were vertically anchored above the metal atoms, which will be discussed later. In addition, the intensity distribution curve (Fig. 1g) from Fig. 1f also confirms the successful incorporation of Au single atoms[34]. EDS elemental analysis shows that Au atoms were uniformly dispersed in MnFeCoNiCu LDH nanosheet (Fig. 1h, Fig. S3b and Table S1). The mass content of Au single atoms measured by inductively coupled plasma-mass spectrometry (ICP-MS) was 1.1 wt.%.

### Resolution of fine coordination environment
To investigate the chemical state of MnFeCoNiCu LDH and $Au_{SA}$-MnFeCoNiCu LDH, XPS analysis was conducted[35]. The high-resolution XPS spectra of all elements in samples were examined, revealing that the Mn, Fe, Co, Ni and Cu elements in $Au_{SA}$-MnFeCoNiCu LDH all shifted to higher binding energy than those in pristine MnFeCoNiCu LDH (Fig. S5). This shift suggests an enhanced chemical valence for metallic ions after the incorporation of Au single atoms[19]. Increasing the metal valence will enhance the orbital hybridization between the metal 3d and O 2p orbitals, resulting in a strengthened M-O covalent bond that favors the LOM rather than AEM in terms of the OER mechanism[7]. For atomically dispersed Au in $Au_{SA}$-MnFeCoNiCu LDH, the high-resolution Au 4f spectrum (Fig. S5f) can be deconvoluted into two characteristic peaks at 87.39 and 83.69 eV, which are lower than the standard peak positions of Au (dotted line), implying a lower valence state. XPS results demonstrate a strong electronic coupling between Au single atoms and MnFeCoNiCu LDH[36]. Additionally, we further analyzed the XPS spectra of O 1s (Fig. 2a) and identified a peak

of 531.0 eV corresponding to oxygen vacancies ($V_O$), which confirms the existence of $V_O$ in $Au_{SA}$-MnFeCoNiCu LDH. This conclusion was further supported by the EPR test result (Fig. S6).

To further study the electronic structure and coordination structure of $Au_{SA}$-MnFeCoNiCu LDH, the hard XAFS measurement was performed. Compared with those in MnFeCoNiCu LDH, all the absorption edges of Mn, Fe, Co, Ni and Cu in $Au_{SA}$-MnFeCoNiCu LDH shift to higher energies (Fig. S7), implying an increased valence of 3d transition metals, which is consistent with the XPS result[34]. Moreover, the decreased intensity of the white line peak on the Au $L_3$-edge of $Au_{SA}$-MnFeCoNiCu LDH indicates a reduced oxidation state of Au (Fig. 2b), which also agrees well with the XPS result. To investigate the local coordination geometry of transition metals in the samples, the Fourier transformed extended X-ray absorption structure measurement was conducted. It was found that the peak intensity of the Ni-O bond became significantly weaker after the introduction of Au (Fig. 2c), while the changes in Mn-O, Fe-O, Co-O and Cu-O were almost negligible (Fig. S8), suggesting the existence of oxygen vacancies near the Ni site, which is coinciding with the XPS result. The fitted curve of the Ni $K$-edge EXAFS (Fig. 2c) indicates that the coordination number (CN) of Ni-O bond in $Au_{SA}$-MnFeCoNiCu LDH is 5, in contrast to the CN (6) of Ni-O bond in MnFeCoNiCu LDH, implying that the oxygen vacancies will also be generated during electrochemical deposition process, which is well consistent with the XPS and EPR results.

Furthermore, as shown in Fig. 2d, $Au_{SA}$-MnFeCoNiCu LDH exhibits a distinct Au-O coordination and no characteristic peak of Au-Au bond

(2.57 Å) is observed[37], which verifies the presence of atomically dispersed Au atoms. The wavelet transform plot for the Au EXAFS $k^2\chi(k)$ of $Au_{SA}$-MnFeCoNiCu LDH shows a maximal intensity of 6.56 Å$^{-1}$, which is contributed by the Au-O scattering (Fig. 2e)[38,39]. The combination of the AC-HAADF-STEM (Fig. 1f) and EXAFS (Fig. 2d) results suggest that there exist two possible coordination geometries for Au atoms. One possible configuration is that the Au atom is anchored on the surface of MnFeCoNiCu LDH, bonding with four oxygen atoms, among which three oxygen atoms are located at the superficial lattice and one is located over the Au atom, as illustrated in Fig. S9a. However, the fitted result based on the above configuration is significantly different from the experimental data (Fig. S9b), implying that Au atoms are not anchoring on the surface. The other possible coordination geometry is for the Au atom is that it may replace a superficial metal site, as shown in Fig. S10. Given that the Au atom could potentially replace any of the metal atoms in MnFeCoNiCu LDH, we calculated the substitution formation energies of $Au_M$ (M = Mn, Fe, Co, Ni, and Cu) via DFT. The calculated result (Fig. S11) indicates that the Au atom prefers to occupy the superficial Fe site in terms of thermodynamics. Therefore, we constructed a structure model in which the Au atom occupies the Fe site in MnFeCoNiCu LDH and then fitted the EXAFS curve (Fig. S12 and Fig. 2d). Notably, the fitted result (Table S2) matches the experimental data well, confirming the doping of a single Au atom at the superficial Fe site.

To further uncover how the incorporated Au single atom occupies the Fe site experimentally, we conducted the ICP-MS test in the

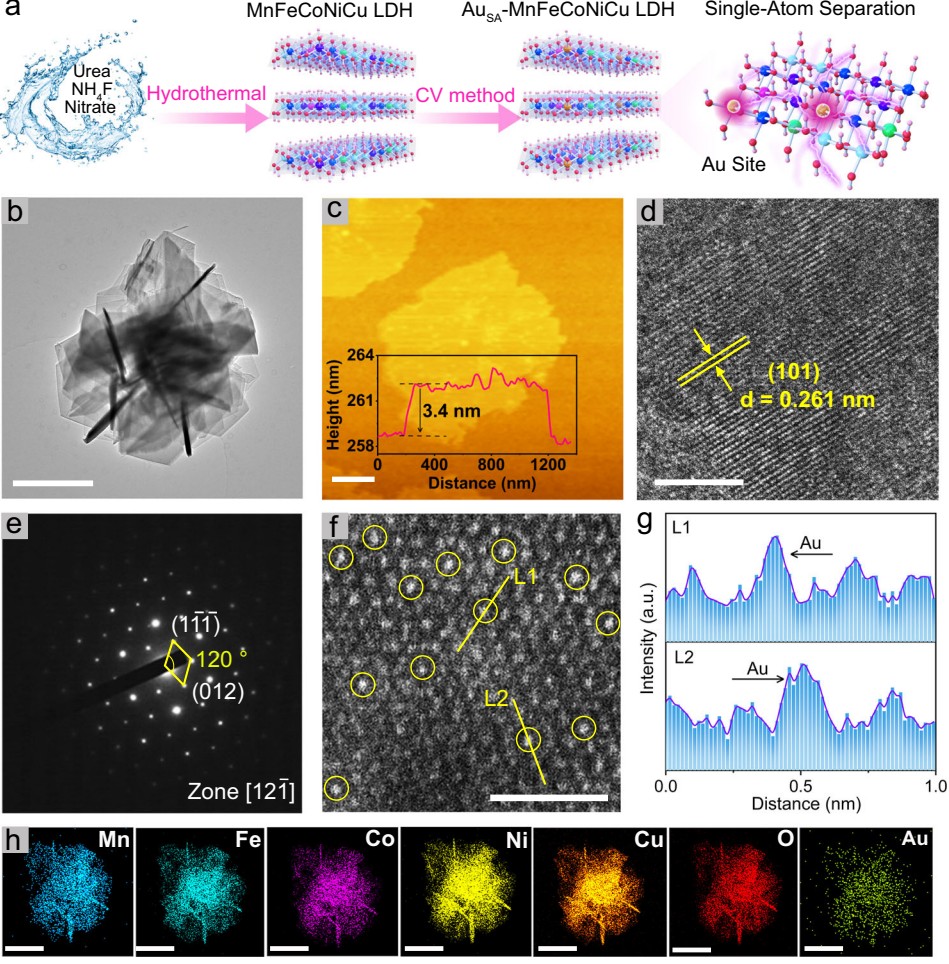

**Fig. 1 | Synthesis scheme and structural characterization of $Au_{SA}$-MnFeCoNiCu LDH. a** Synthesis schematic of $Au_{SA}$-MnFeCoNiCu LDH. **b** TEM image of $Au_{SA}$-MnFeCoNiCu LDH. The scale bar is 1 μm. **c** AFM image. The scale bar is 0.3 μm. **d** HRTEM image. The scale bar is 10 nm. **e** SAED pattern. **f** AC-HAADF-STEM image. The scale bar is 1 nm. **g** The corresponding intensity profiles of L1 and L2 in (**f**). **h** EDS elemental mapping of Mn, Fe, Co, Ni, Cu, O, Au. The scale bar is 1 μm.

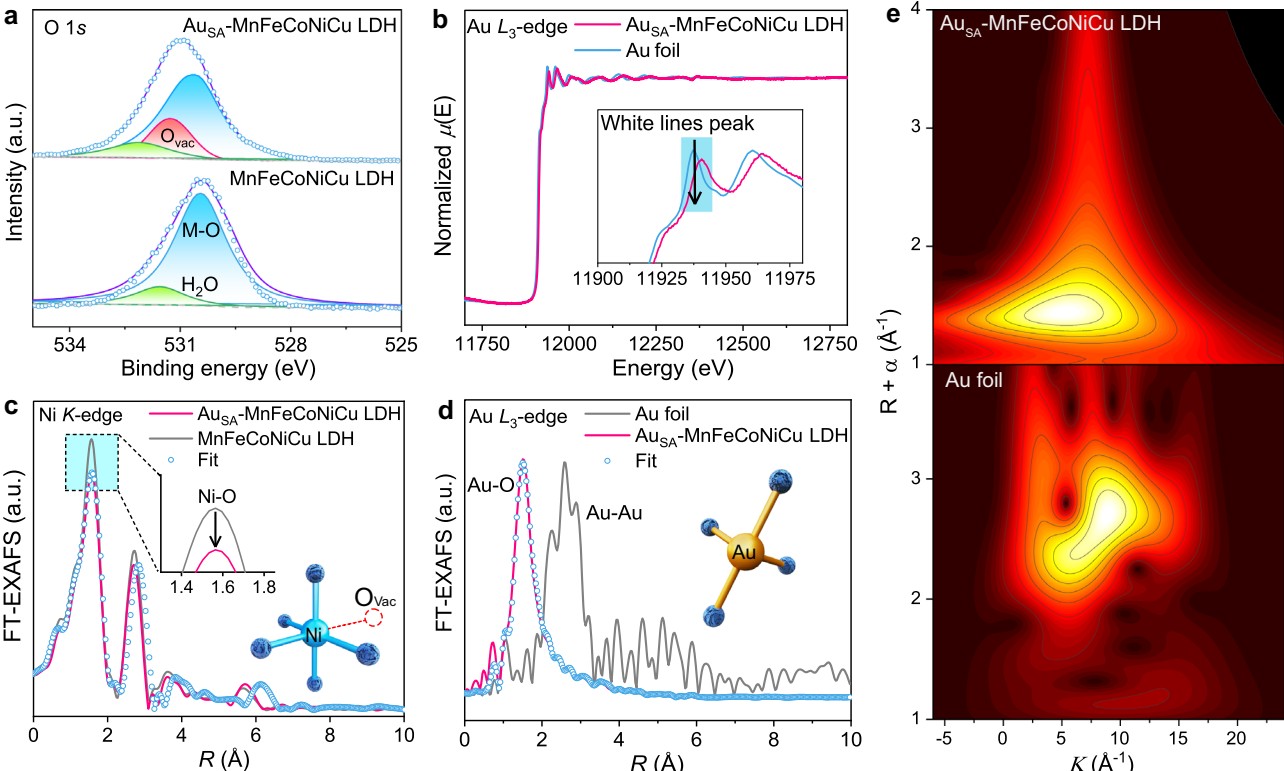

**Fig. 2 | Chemical coordination of Au$_{SA}$-MnFeCoNiCu LDH. a** O 1$s$ XPS spectra of Au$_{SA}$-MnFeCoNiCu LDH and MnFeCoNiCu LDH. **b** Normalized Au $L_3$-edge XANES spectra of Au$_{SA}$-MnFeCoNiCu LDH and Au foil. (The inset is an enlarged view of the white line peak.) **c** Ni $K$-edge EXAFS k$^2$χ(k) Fourier transform (FT) spectra of Au$_{SA}$-MnFeCoNiCu LDH and MnFeCoNiCu LDH, where the insert is the fitting model (blue for Nickel, navy for Oxygen). **d** EXAFS k$^2$χ(k) Fourier transform (FT) spectra of Au$_{SA}$-MnFeCoNiCu LDH and Au foil (model: golden yellow for Gold, navy for Oxygen). **e** Wavelet transform (WT) contour map for EXAFS k$^2$χ(k) of Au$_{SA}$-MnFeCoNiCu LDH and Au foil.

electrolyte after CV electrochemical deposition and on the electrode before CV. The result indicates that the dissolution percentage of Fe ions is far more than that of other metals (Fig. S13), which also means the abundant Fe vacancies on the surface of MnFeCoNiCu LDH. This experimental finding is supported by DFT calculations on the formation energies of metal vacancies in MnFeCoNiCu LDH (Fig. S14). Hence, it is reasonable to assume that the incorporated Au single atom tends to occupy the Fe site due to the abundant Fe vacancies on the surface of the catalyst. Finally, by combining the XPS and XAFS analysis, we have verified the successfully incorporation of Au single atoms and oxygen vacancies in high-entropy MnFeCoNiCu LDH, and the enhanced valence of the transition metals in high-entropy LDH caused by the strong electronic coupling may alter the OER mechanism[40].

## Electrocatalytic performance toward oxygen evolution

After determining the geometric and electronic structure of Au$_{SA}$-MnFeCoNiCu LDH, we further performed the electrochemical test towards OER in 1.0 M KOH using a standard three-electrode system. Before the OER testing, electrolyte was purified to eliminate the impact of trace amounts of Fe according to previously reported methods[41]. The polarization curve of Au$_{SA}$-MnFeCoNiCu LDH with 95% iR correction exhibits a significantly boosted OER activity compared to MnFeCoNiCu LDH and commercial IrO$_2$ (Fig. 3a). For Au$_{SA}$-MnFeCoNiCu LDH, it only requires a low overpotential (η) of 213, 260, and 263 mV to reach current densities of 10, 100, and 250 mA cm$^{-2}$, respectively. These values are 110, 183, and 229 mV lower than those of pristine MnFeCoNiCu LDH (Fig. 3b). We also carried out LSV tests for Au$_{SA}$-MnFeCoNiCu LDH and MnFeCoNiCu LDH with different iR compensations (Fig. 3a and Fig. S15), which are discussed in Supplementary note 1. To further investigate the OER reaction kinetics, the Tafel

slopes were extracted from the polarization curve (Fig. 3c). Compared with that of MnFeCoNiCu LDH (85.5 mV dec$^{-1}$) and IrO$_2$ (59.1 mV dec$^{-1}$), the Tafel slope of Au$_{SA}$-MnFeCoNiCu LDH is dramatically reduced to 27.5 mV dec$^{-1}$, which not only indicates a faster reaction kinetics, but also implies a possible change in the OER reaction mechanism[42]. Besides Au$_{SA}$-MnFeCoNiCu LDH, we also successfully synthesized 20 Au-decorated high-entropy LDHs materials (Au$_{SA}$-HE LDHs) with different quinary transition-metals composition (Cr, Mn, Fe, Co, Ni Cu or Zn), and tested their OER performance (Figs. S16–S20 and Table S3), as shown in Supplementary note 2. Considering the low η and the small Tafel slope, the as-synthesized Au$_{SA}$-MnFeCoNiCu LDH exhibits better OER activity than other Au$_{SA}$-HE LDHs in this works (Fig. S18) and most reported high-entropy electrocatalysts (Fig. 3d)[19,20,43–49].

Additionally, the kinetics difference of MnFeCoNiCu LDH and Au$_{SA}$-MnFeCoNiCu LDH was further verified by the electrochemical impedance spectroscopy (EIS) test. The Nyquist plot with an equivalent circuit is shown in Fig. 3e, where Au$_{SA}$-MnFeCoNiCu LDH delivers a smaller charge-transfer resistance (R$_{ct}$) value (1.1 Ω) than MnFeCoNiCu LDH (9.8 Ω). This suggests a much faster charge transfer at the solid-liquid interface using Au$_{SA}$-MnFeCoNiCu LDH, which is beneficial for boosting the OER kinetics[7,50]. Besides, the wettability of the electrode is also crucial as the rapid generation of bubbles at high current densities may block the active sites and impede the mass transfer. As shown in Fig. S21, the droplet contact angle of Au$_{SA}$-MnFeCoNiCu LDH is 22°, significantly smaller than that of MnFeCoNiCu LDH (51.9°), implying better hydrophilicity. The hydrophilic property is beneficial for eliminating gas bubbles and facilitating mass transfer, especially under large current density conditions[51,52].

To further compare the intrinsic activity of different electrocatalysts, the polarization curves were normalized by the

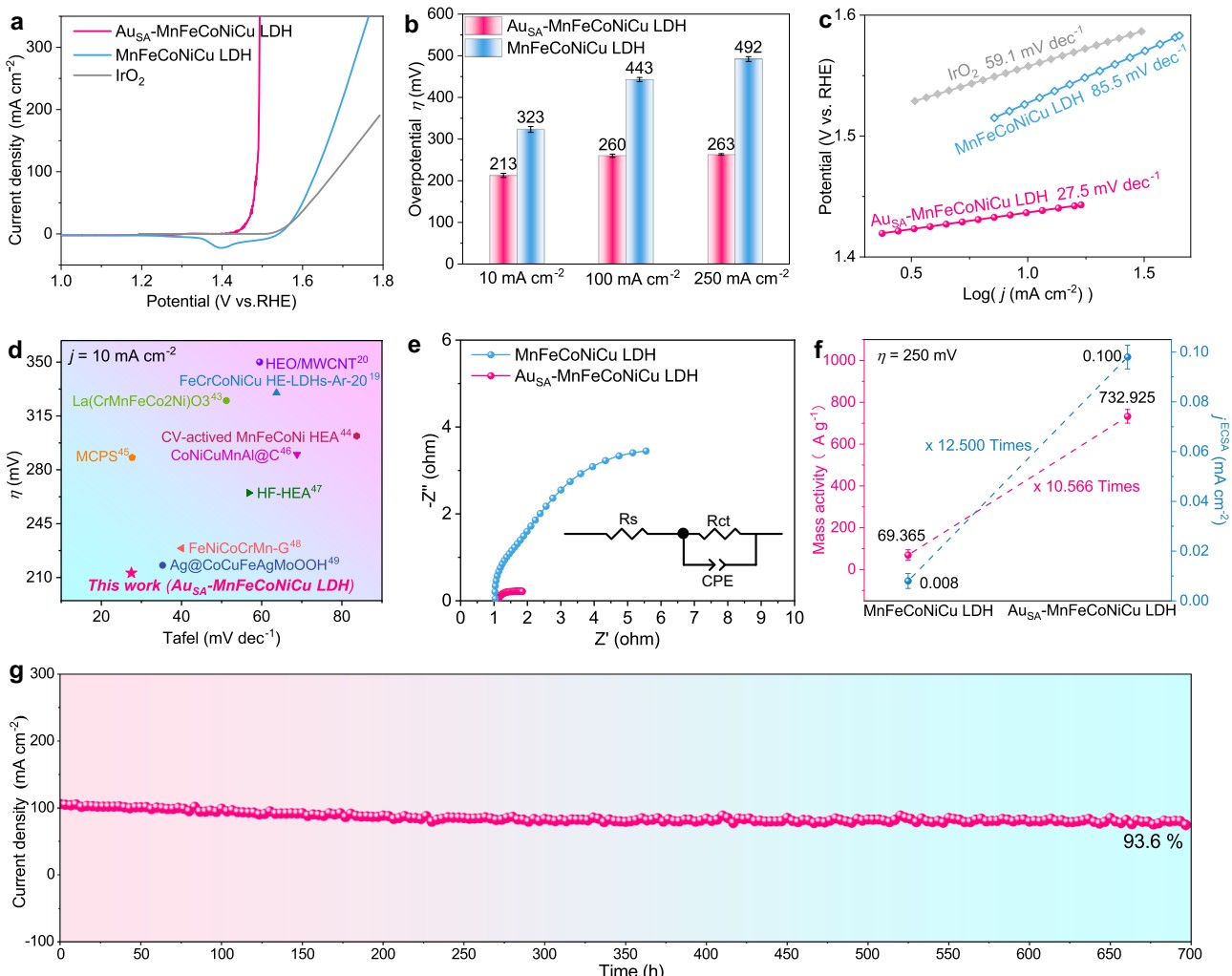

**Fig. 3 | Electrocatalytic performance evaluation of Au$_{SA}$-MnFeCoNiCu LDH in 1 M KOH (pH = 14). a** LSV curves (the scan rate is 2 mV s$^{-1}$) with 95% iR correction. The loading of catalysts is ~1 mg cm$^{-2}$, and the solution resistance is 2.0 Ω. **b** Overpotential comparison (error bar: standard error of three repeated measurements). **c** Tafel plots of Au$_{SA}$-MnFeCoNiCu LDH, MnFeCoNiCu LDH and IrO$_2$. **d** OER performance comparison between Au$_{SA}$-MnFeCoNiCu LDH and other reported high-entropy materials at *j* = 10 mA cm$^{-2}$. **e** EIS curves (the applied potential is 0.459 V vs. Hg/HgO, and the frequency range is 10$^6$–10$^{-2}$ Hz) of Au$_{SA}$-MnFeCoNiCu LDH and MnFeCoNiCu LDH (inset is equivalent circuit). **f** Mass activity and normalized (by ECSA) current density of Au$_{SA}$-MnFeCoNiCu LDH and MnFeCoNiCu LDH at η = 250 mV (error bar: standard error of three repeated measurements). **g** Stability test of Au$_{SA}$-MnFeCoNiCu LDH.

electrochemical active surface area (ECSA), where the ECSA values were calculated by the electric double-layer capacitance (C$_{dl}$) (Fig. S22a), as described in the experimental section[53]. The ECSA-normalized LSV curves of Au$_{SA}$-MnFeCoNiCu LDH and MnFeCoNiCu LDH (Fig. S22b) demonstrate that the intrinsic activity of Au$_{SA}$-MnFeCoNiCu LDH is still superior to that of MnFeCoNiCu LDH. At η = 250 mV, Au$_{SA}$-MnFeCoNiCu LDH shows a specific activity of 0.100 mA cm$^{-2}$, which is 12.500 times higher than that of MnFeCoNiCu LDH (0.008 mA cm$^{-2}$). Moreover, the mass activity of Au$_{SA}$-MnFeCoNiCu LDH (732.925 A g$^{-1}$) at an overpotential of 250 mV is 10.566 times higher than that of MnFeCoNiCu LDH (69.365 A g$^{-1}$) (Fig. 3f and Table S4). All these results prove that the incorporation of Au single atoms and oxygen vacancies could significantly improve the intrinsic activity of MnFeCoNiCu LDH. It is noteworthy that, in addition to the activity, stability is also a crucial factor for most electrocatalysts. Encouragingly, the synthesized Au$_{SA}$-MnFeCoNiCu LDH shows good stability for OER. Even after a 700 h chronoamperometry test (I-t) at a constant potential of 1.53 V (vs. RHE), the current density (~100 mA cm$^{-2}$) only decays by 6.4% (Fig. 3g), suggesting significantly better long-term stability than most LDHs and oxyhydroxides OER catalysts (Table S5).

It is also noteworthy that the activity decay is primarily concentrated in the initial 50 h, with an activity decay of 5.7% (Fig. S23), so we conclude the high-entropy catalyst may undergo a surface reconstruction process during the early stage. Hence, the structure characterization of Au$_{SA}$-MnFeCoNiCu LDH after 50 h I-t test was carried out. The XRD results (Fig. S24) show that Au$_{SA}$-MnFeCoNiCu LDH still maintains its initial crystal structure. The HRTEM result (Fig. S25) validates the generation of an amorphous layer with a thickness of about 10 nm, which may be an amorphous oxyhydroxides[36,54]. To further identify the amorphous layer, an in-situ Raman spectrum test was performed, as illustrated in Fig. S26, which confirms the formation of amorphous oxyhydroxides. Moreover, the element mapping analysis (Fig. S27) also shows that Mn, Fe, Co, Ni, Cu, O, and Au elements are still uniformly distributed in the catalyst, suggesting the maintenance of the high-entropy structure. These results indicate that the surface of the high-entropy LDH catalyst undergoes a surface reconstruction during OER, i.e., transforming from high-entropy LDH to high-entropy oxyhydroxides. Further extending the I-t test time up to 700 h had little impact on the structure of the high-entropy catalyst, as shown in Figs. S28–S30, suggesting the good structure stability for our as-synthesized high-entropy catalyst after the initial surface

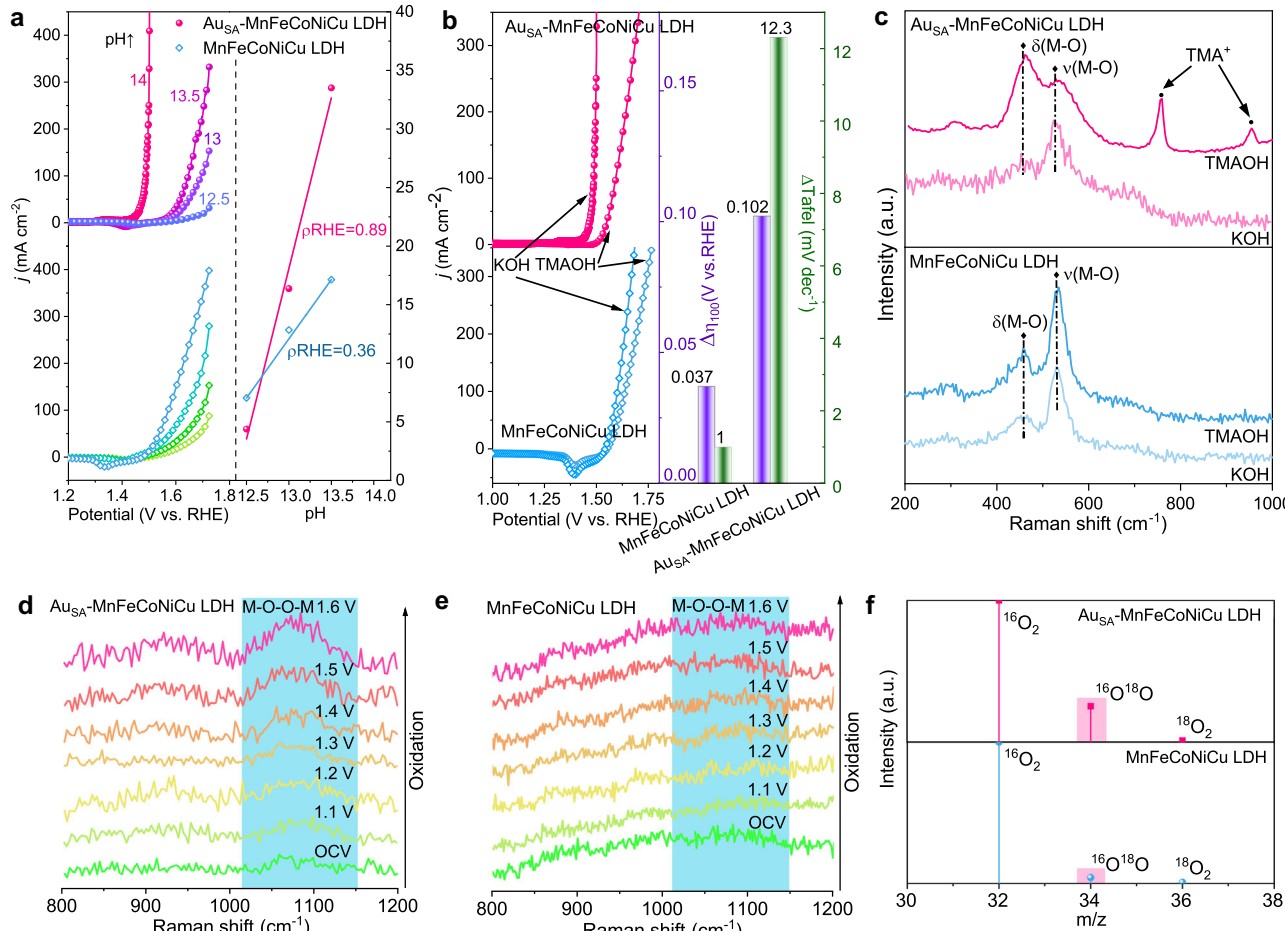

**Fig. 4 | Electrochemical and spectroscopic verification of LOM. a** LSV curves measured in KOH electrolytes with pH = 12.5, 13, 13.5, and 14 (left), $j$ at 1.45 V vs. RHE plotted in log scale as a function of pH (right), from which the proton reaction orders ($\rho^{RHE} = \partial log j/\partial pH$) were derived. The loading of catalysts is -1 mg cm$^{-2}$, and the solution resistance is 2.0 Ω. **b** Polarization curves of Au$_{SA}$-MnFeCoNiCu LDH and MnFeCoNiCu LDH in 1.0 M KOH and 1.0 M TMAOH (left), shift of overpotential at 100 mA cm$^{-2}$ ($\Delta\eta_{100}$) and Tafel slopes from KOH to TMAOH (right). **c** Raman spectra of Au$_{SA}$-MnFeCoNiCu LDH and MnFeCoNiCu LDH, measured after running at 1.45 V vs. RHE for 30 min in 1.0 M TMAOH and 1.0 M KOH solutions and washing with deionized water. **d** In situ Raman spectra of MnFeCoNiCu LDH. **e** In situ Raman spectra of Au$_{SA}$-MnFeCoNiCu LDH. **f** Mass spectrometric results by $^{18}$O isotope-labeling experiments. The signals were normalized by initial intensity of $^{16}$O$_2$.

reconstruction. Additionally, the AC-HAADF-STEM and XPS results of the high-entropy catalyst after the stability test also demonstrate the existence of Au single atoms and O vacancies. (Fig. S31)

Recent studies have shown that the dissolution of Fe species in Fe-based LDH or oxyhydroxides can lead to the OER performance decay[41,55]. In this study, our high-entropy catalyst exhibited robust stability, we thus hypothesize that the dissolution of Fe species in our high-entropy catalyst might have been inhibited. To validate this hypothesis, we conducted the ICP-MS test on the electrolyte after the stability test to quantify the dissolution percentage of metal ions. Stability tests were also performed on NiFe LDH as a comparison (Fig. S32). As shown in Fig. S33, the dissolution percentage of Fe ions for Au$_{SA}$-MnFeCoNiCu LDH is only 3.5%, obviously lower than that for NiFe LDH (30%), supporting our hypothesis that the unique high-entropy effect and sluggish diffusion effect of HEMs contribute to the catalyst's stability[56]. Moreover, the HER performance of Au$_{SA}$-MnFeCoNiCu LDH and MnFeCoNiCu LDH were also explored, see Supplementary note 3 and Fig. S34.

**Identification of OER mechanism**

Noting the dramatic reduction in the Tafel slope of Au$_{SA}$-MnFeCoNiCu LDH relative to MnFeCoNiCu LDH (Fig. 3c) indicates a substantial change in reaction kinetics, which could be attributed to the variation of OER mechanism. To verify this conjecture, we conducted the LSV

measurement under different pH values ranging from 12.5 to 14.0[11,57]. As illustrated in Fig. 4a, it is found that the OER activity of Au$_{SA}$-MnFeCoNiCu LDH enhances significantly with increasing pH values, while MnFeCoNiCu LDH exhibits slight pH-dependent activity. To precisely clarify the correlation between the activity and pH values, the proton reaction orders on RHE scale ($\rho^{RHE}$, $\rho^{RHE} = \partial log(j)/\partial pH$) was used, which reflects the dependence of OER reaction kinetics on proton activity. The detailed calculation process was described in the electrochemical measurements section[58,59]. When this value is closer to 1, the pH-dependent property of the catalyst is stronger[60]. Considering that the $\rho^{RHE}$ value of Au$_{SA}$-MnFeCoNiCu LDH (0.89) is closer to 1 comparing with that of MnFeCoNiCu LDH (0.36), Au$_{SA}$-MnFeCoNiCu LDH shows a stronger pH-dependent property, implying that it may undergo LOM rather than the traditional AEM during OER[61].

Unlike AEM, the LOM will generate O$_2$$^{2-}$ species during OER, so detecting O$_2$$^{2-}$ species is important for verifying the OER mechanism[11]. Therefore, tetramethylammonium cation (TMA$^+$), which can strongly bind to O$_2$$^{2-}$ species and hinder OER kinetics, was introduced as a detector of O$_2$$^{2-}$ species[58]. As shown in Fig. 4b, Au$_{SA}$-MnFeCoNiCu LDH shows significantly reduced OER activity in 1.0 M TMAOH electrolyte relative to 1.0 M KOH electrolyte ($\Delta\eta_{100}$ = 0.102 V, $\Delta$Tafel = 12.3 1 mV dec$^{-1}$), implying the binding between TMA$^+$ and O$_2$$^{2-}$ and validating that Au$_{SA}$-MnFeCoNiCu LDH undergoes LOM during OER. Inversely, MnFeCoNiCu LDH exhibits a

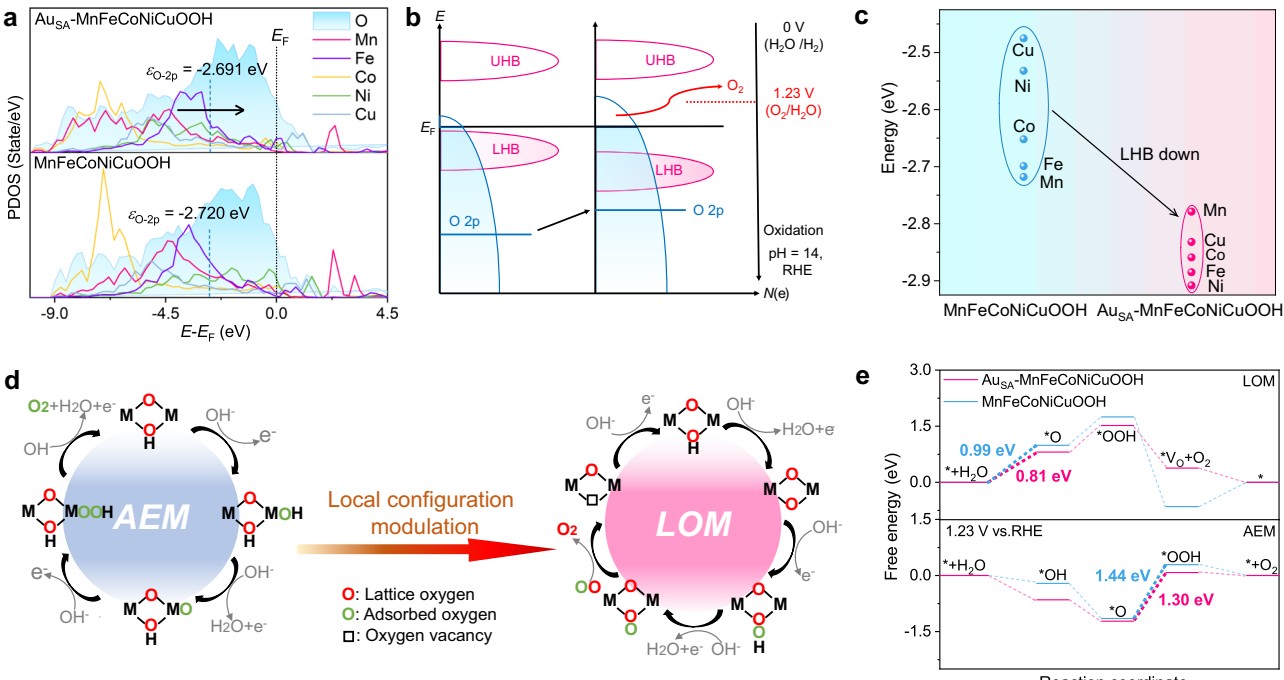

**Fig. 5 | Analysis of molecular orbital and reaction kinetics. a** Projected density of states ($E_F$: Fermi level, $\varepsilon_{O-2p}$: O $2p$ band center); **b** Schematic band diagrams (UHB: upper Hubbard band, LHB: lower Hubbard band, $N(e)$: state density); **c** The LHB center positions of $Au_{SA}$-MnFeCoNiCuOOH and MnFeCoNiCuOOH. **d** Adsorbate

evolution mechanism (AEM) and oxygen oxidation mechanism (LOM) on MnFe-CoNiCuOOH. **e** Computed free energies (ΔG) of OER steps on $Au_{SA}$-MnFeCoNi-CuOOH and MnFeCoNiCuOOH.

slight change ($\Delta\eta_{100} = 0.037\,V$, $\Delta Tafel = 1\,mV\,dec^{-1}$), indicating an AEM pathway.

Raman spectroscopy was also leveraged to validate the presence of $O_2^{2-}$ species (Fig. 4c). MnFeCoNiCu LDH and $Au_{SA}$-MnFeCoNiCu LDH were operated in 1.0 M TMAOH and 1.0 M KOH solutions at 1.45 V vs. RHE for 30 min, respectively, and then washed with deionized water prior to the Raman spectroscopy measurement. The $Au_{SA}$-MnFeCoNiCu LDH exhibits two peaks at 751.73 and 950.54 $cm^{-1}$, corresponding to the characteristic peaks of TMA$^+$[7]. In contrast, no characteristic peaks are observed in MnFeCoNiCu LDH, validating the existence of $O_2^{2-}$ species in $Au_{SA}$-MnFeCoNiCu LDH during OER[7,58]. The two major Raman peaks located at 400–600 $cm^{-1}$ (Fig. 4c) are assigned to the $E_g$ bending vibration of M-O (δ(M-O)) and $A_g$ stretching vibration (v(M-O))[40,62]. Furthermore, the presence of $O_2^{2-}$ was directly verified by in situ electrochemical Raman spectroscopy (Fig. 4d). For $Au_{SA}$-MnFeCoNiCu LDH, when the applied potential reaches 1.3 V (vs. RHE), a broad Raman peak around 1089 $cm^{-1}$ can be observed, which is ascribed to the stretching vibration of $O_2^{2-}$ (*-O-O-*) species[60]. As the applied potential increases, the characteristic Raman peaks of $O_2^{2-}$ species become stronger and sharper, indicating the substantial generation of $O_2^{2-}$ species and confirming the involvement of LOM during OER. In contrast, this characteristic Raman peak of $O_2^{2-}$ species is tiny for the pristine MnFeCoNiCu LDH (Fig. 4e), implying a dominated AEM pathway[60,63].

The afore-mentioned discussions reveal that $Au_{SA}$-MnFeCoNiCu LDH follows the LOM pathway during OER while MnFeCoNiCu LDH mainly follows the AEM pathway. To further reveal the participation of lattice oxygen in the OER process, the $^{18}O$-labeled gas chromatography-mass spectrometer (GC-MS) measurements were performed, and the test details were described in the experimental section. Both $Au_{SA}$-MnFeCoNiCu LDH and MnFeCoNiCu LDH were firstly activated by electrochemical CV method in $^{18}O$-labeled KOH electrolyte, and then carried out OER test in 1.0 M KOH with regular $H_2O$. The collected oxygen products measured by GC-MS (Fig. 4f) validate the existence of $^{18}O$-labeled products such as $^{16}O^{18}O$ and $^{18}O_2$

for both samples, suggesting that both $Au_{SA}$-MnFeCoNiCu LDH and MnFeCoNiCu LDH involve in LOM. However, the content of $^{16}O^{18}O$ product in $Au_{SA}$-MnFeCoNiCu LDH is significantly higher than that in MnFeCoNiCu LDH, implying that more lattice oxygen in $Au_{SA}$-MnFeCoNiCu LDH was involved in the OER reaction compared with that in MnFeCoNiCu LDH. In other words, $Au_{SA}$-MnFeCoNiCu LDH is more inclined to follow the LOM during OER while MnFeCoNiCu LDH favors the AEM.

Considering the greatest electronegativity of Au (2.54) among all transition metals, we conclude that the high electronegativity of Au may be responsible for the activation of the lattice oxygen in high-entropy LDH, because it may induce high metal-oxygen covalency[26]. To further reveal the relationship between the electronegativity of the incorporated metal atoms and the oxygen evolution mechanism, we choose other three metals with different electronegativity (Ru: 2.2, Pt: 2.3 and Ag: 1.9) for comparison, and the results validate that incorporating the single-atom metal with higher electronegativity is more conductive to triggering LOM in high-entropy LDH (Fig. S35–S39, Table S6), as detailed in Supplementary note 4.

### Theoretical insight into LOM

In order to gain insight into the intrinsic reasons for the OER pathway conversion, we performed DFT + U calculations. Due to the surface reconfiguration of the catalyst in the OER process discussed in the previous section, we selected $Au_{SA}$-MnFeCoNiCuOOH and MnFeCoNiCuOOH instead of LDH as the computational models, and the (100) faces as the active surface due to their high activity[30,64]. To reveal the activity of lattice oxygen in $Au_{SA}$-MnFeCoNiCuOOH and MnFeCoNiCuOOH, the density of states of O $2p$ and M $3d$ orbits were calculated (Fig. 5a). Given that the distance between the O $2p$ band center and the Fermi level ($E_F$) was regarded an important parameter to identify the activity of the lattice oxygen, we calculated the O $2p$ band center ($\varepsilon_{O-2p}$) for $Au_{SA}$-MnFeCoNiCuOOH and MnFeCoNiCuOOH, and the values were −2.691 and −2.720 eV, respectively. The O $2p$ band center after

introducing Au shifts towards $E_F$, promoting the release of the lattice oxygen from the lattice, which will facilitate the LOM[10,65].

According to molecular orbital theory, the bond between M and O in MOOH will lead to the formation of M-O bonding bands with oxygen character and M-O* antibonding bands with M character. For late transition metal oxyhydroxides, the strong d-d Coulomb interaction in M-O* antibonding orbitals will give rise to the Mott-Hubbard splitting, generating an empty upper Hubbard band (UHB) and a lower Hubbard energy band (LHB) full of electrons, as illustrated in Fig. 5b[7,66]. It is noted that the energy difference between the UHB center and the LHB center (ΔU) is also a useful descriptor to evaluate the lattice oxygen activity[67]. A larger ΔU implies a stronger d-d Coulomb interaction, resulting in the LHB penetrating into the M-O bonding band. This enables the electrons to mainly remove from M-O bonding band rather than from the LHB under anodic potential, which will weaken the metal-oxygen bonds due to the decrease of M-O bond order and facilitate the lattice oxygen oxidation[7]. Therefore, we compared the LHB center and ΔU values of $Au_{SA}$-MnFeCoNiCuOOH and MnFeCoNiCuOOH, as shown in Table S7 and Table S8. The DFT + U calculations show that Au incorporation weakens the metal-oxygen bonds in MnFeCoNiCuOOH, as indicated by the increased ΔU values of M-O bands including Mn-O、Fe-O、Co-O、Ni-O and Cu-O bonding bands in MnFeCoNiCuOOH relative to MnFeCoNiCuOOH. This weakened metal-oxygen bonding implies that $Au_{SA}$-MnFeCoNiCuOOH favors the LOM than MnFeCoNiCuOOH. Furthermore, the LHB center value of Ni-O bonding band (−2.908 eV) is the most negative among all M-O LHB centers (Mn-O, −2.779 eV; Fe-O, −2.885 eV; Co-O, −2.859 eV; Cu-O, −2.832 eV) in $Au_{SA}$-MnFeCoNiCuOOH, suggesting the weakest Ni-O bond (Fig. 5c). In a word, $Au_{SA}$-MnFeCoNiCuOOH possesses the upshifted O 2p band, the increased ΔU value, and the weakened M-O bond compared with MnFeCoNiCuOOH, implying a higher propensity for triggering the lattice oxygen activation and a preference for LOM, agreeing well with the experimental result[7,67].

In general, AEM consists of four basic steps with three different intermediates (*OH, *O, *OOH), while LOM involves five basic steps including four different intermediates (*O, *OOH, *OO, $V_O$), as shown in Fig. 5d. Based on the AEM and LOM pathways, the Gibbs adsorption free energy diagrams of $Au_{SA}$-MnFeCoNiCuOOH and MnFeCoNiCuOOH were calculated. Firstly, we calculated the adsorption free energy of OER intermediates on different transition metal sites in $Au_{SA}$-MnFeCoNiCuOOH and MnFeCoNiCuOOH (Fig. 5e, Figs. S40−S43 and Table S9). The results indicate that Ni sites exhibit the lowest energy barrier (1.30 eV and 1.44 eV) for both samples in the AEM pathway. For comparison, the Gibbs free energy profiles of $Au_{SA}$-MnFeCoNiCuOOH and MnFeCoNiCuOOH following LOM are also provided (Fig. 5e, Figs. S44, S45 and Table S10). Here, the oxygen atoms coordinated with Ni sites were chosen as the active sites due to the weakest Ni-O bond among all M-O bonds mentioned above. It can be observed that the first electrochemical deprotonation steps in the LOM pathway are the rate-determining steps for both $Au_{SA}$-MnFeCoNiCuOOH and MnFeCoNiCuOOH, with energy barriers of 0.81 eV and 0.99 eV, respectively. From the perspective of thermodynamics, $Au_{SA}$-MnFeCoNiCuOOH prefers to follow LOM relative to MnFeCoNiCuOOH, which is well consistent with the experimental results.

To further reveal the synergistic effect between the Au single atoms and the oxygen vacancies in $Au_{SA}$-MnFeCoNiCuOOH, we also calculated the Gibbs free energy profiles of MnFeCoNiCuOOH with only Au single atom and MnFeCoNiCuOOH with only oxygen vacancy following the LOM pathway, as illustrated in Figs. S46−S48 and Table S10. The LOM energy barriers for MnFeCoNiCuOOH with only Au atom and MnFeCoNiCuOOH with only O vacancy are 1.23 and 0.95 eV, respectively, both of which are larger than that of $Au_{SA}$-MnFeCoNiCuOOH (0.81 eV). This further confirms the synergistic effect of Au single atoms and O vacancies in triggering the LOM in high-entropy MnFeCoNiCuOOH. In order to elucidate the effect of high entropy on

the OER process from the viewpoint of theoretical studies, we constructed a high-entropy $Au_{SA}$-MnFeCoNiCuOOH model with disordered 3d transition metals and a non-high-entropy $Au_{SA}$-MnFeCoNiCuOOH model with ordered 3d transition metals, as illustrated in Fig. S49. We calculate the formation energies of the high-entropy and non-high-entropy samples, respectively, and the results show that the formation energy of the high-entropy sample (−1.75 eV) is significantly lower than that of the non-high-entropy sample (−0.21 eV), suggesting that the high-entropy effect can strengthen the structure stability of the catalyst (Fig. S50). Moreover, given that the easy leaching of metal species next to oxygen vacancies in LOM-based electrocatalysts during OER[10,68], we also calculated the binding energies of Ni species next to oxygen vacancies in high-entropy and non-high-entropy samples, which indicates that the binding energy of Ni next to oxygen vacancies in high-entropy sample (−1.35 eV) is more negative than that of non-high-entropy sample (−0.25 eV), implying that high-entropy effect can inhibit the leaching of metal species and boost the structure stability during lattice-oxygen oxidation process, which is responsible for the good OER stability (Fig. S51).

## Discussion

In summary, we report a high-entropy MnFeCoNiCu LDH decorated with Au single atoms and O vacancies, which is fabricated by hydrothermal and electrochemical deposition methods. This catalyst delivers a remarkable enhancement in OER activity compared to MnFeCoNiCu LDH, and display a superior long-term durability. The [18]O isotope labeling mass spectroscopy in combination with ex/in situ Raman spectroscopy validates that the boosted activity is attributed to the transformation of OER mechanism from AEM to LOM. DFT + U calculations further confirm that the introduced Au single atoms and the oxygen vacancies can synergistically upshift O 2p orbits and weaken metal-oxygen bonds to activate the lattice oxygen and lower the energy barrier of LOM, which facilitates the lattice oxygen oxidation. Moreover, the high-entropy effect is responsible for the good OER stability. This work provides valuable insights for designing robust and stable high-entropy electrocatalysts for a host of catalytic reactions involving lattice oxygen.

## Methods

**Materials.** $Ni(NO_3)_2 \cdot 6H_2O$, $Fe(NO_3)_2 \cdot 9H_2O$, $Co(NO_3)_2 \cdot 6H_2O$, $Mn(NO_3)_2 \cdot 4H_2O$, $Cu(NO_3)_2 \cdot 3H_2O$ and $HAuCl_4$ were obtained from Sigma−Aldrich. $NH_4F$ and urea were obtained from Macklin.

**MnFeCoNiCu LDH preparation.** MnFeCoNiCu LDH was prepared by a hydrothermal method. First, $Ni(NO_3)_2 \cdot 6H_2O$ (0.45 mmol), $Fe(NO_3)_2 \cdot 9H_2O$ (0.45 mmol), $Co(NO_3)_2 \cdot 6H_2O$ (0.45 mmol), $Mn(NO_3)_2 \cdot 4H_2O$ (0.45 mmol), and $Cu(NO_3)_2 \cdot 3H_2O$ (0.45 mmol) were dissolved in 35 mL of deionized water, and then $NH_4F$ (4 mmol) and urea (10 mmol) were added. After complete dissolution, the mixed solution was poured into an autoclave, and added a piece of treated carbon cloth (CC) inside also, followed by heating to 120 °C for 6 h.

**Au single-atom separation method.** The Au single-atom loaded MnFeCoNiCu LDH was further prepared using the electrochemical CV method. The electrolyte used was a mixture of NaCl and $HAuCl_4$ (4 mmol), mixed in 100 mL of deionized water. The working electrode was prepared MnFeCoNiCu LDH, the counter electrode was carbon rod, and the reference electrode was Hg/HgO. Then in the voltage range of −0.6 to −0.2 V vs. RHE, conduct 5 CV cycles with scan rate of 0.1 mV s$^{-1}$. After finishing, the working electrode was rinsed with deionized water and dried. The obtained $Au_{SA}$-MnFeCoNiCu LDH was then washed with deionized water and dried.

**Characterization.** Crystal structure characterizations were performed using an automated multipurpose X-ray diffractometer (SmartLab, Rigaku), and the employed X-ray source is 4 kW rotating anode Cu-Kα, with a wavelength of 1.54059 Å. The morphology of the catalysts was captured by the focused ion beam scanning electron

microscope (FIB-SEM, GAIA3, TESCAN) and the field emission electron microscope (JEM-2100F, JEOL). STEM-HAADF images were obtained from an atomic resolution electron microscope (JEM-ARM300F GRAND ARM, JEOL) with an additional EDS. AFM imaging was performed on the scanning electron probe microscope (5500 AFM/STM, Agilent). XPS were measured on the X-ray photoelectron spectrometer (VersaProbe II, ULVAC-PHI) with an aluminum anode X-ray source, and all spectra were calibrated by using the binding energy of C $1s$ (284.8 eV) as a reference. The Raman spectra were gathered using the Raman spectrometer (LabRAM HR Evolution, HORIBA France SAS) with the excitation wavelength set to 532 nm. The X-ray absorption spectroscopy were recorded at BL14B2 beamline of the SPring-8 Synchrotron Radiation Facility.

Electrochemical measurements. To eliminate the impact of trace iron ions in the electrolyte on the performance of OER, the KOH electrolyte was purified according to literature methods[34] before the OER testing. In addition, we conducted Hg/HgO electrode calibration before the testing, during which platinum wire was used as the working/counter electrode: high-purity hydrogen was bubbled into the 1.0 M KOH electrolyte for 30 min at room temperature firstly, and then the CV method is used for scanning with the range of −1.2–0 V vs. Hg/HgO and the speed of 1 mV s$^{-1}$. Using the CHI-604E electrochemical station, electrochemical measurements were made on the catalysts in the three-electrode system. Except when otherwise noted, electrochemical experiments were performed with an electrolyte of 1.0 M aqueous KOH solution. The catalysts supported on CC was used as the working electrode, while the reference electrode and the counter electrode were the same as above. CV curves were collected at the scan rates of 100 mV s$^{-1}$ and 20 mV s$^{-1}$, and the linear sweep voltammetry (LSV) were measured at 2 mV s$^{-1}$ (95% iR corrected). The solution resistance is 2.0 Ω. Double-layer capacitance ($C_{dl}$) was obtained from CV at different scan rates. The ECSA of the ample was derived from the $C_{dl}$ according to the following equation:

$$ECSA = C_{dl}/C_S \tag{1}$$

where $C_S$ get a value of 40 μF cm$^{-2}$. EIS was performed at 0.45926 V vs. Hg/HgO, the amplitude and the frequency range were set to 10 mV and $10^6$–$10^{-2}$ Hz, respectively. Chronoamperometry tests were performed at a steady voltage of 1.5 V vs. RHE to evaluate the stability.

Proton reaction order ($\rho^{RHE}$) reflects the dependence of OER reaction kinetics on proton activity, and the formula is as follows.

$$\rho^{RHE} = \partial \log(j)/\partial pH \tag{2}$$

Where the pH value ranges from 12.5 to 14 and log(j) is the logarithm of the current density at 1.45 V vs. RHE. When proton coupled electron transfer reactions occur, the OER kinetics are almost independent on the pH value of the solution, resulting in a low $\rho^{RHE}$. If non synergistic proton electron transfer is involved in OER, the OER kinetics would be strongly pH-dependent and have a large $\rho^{RHE}$ value.

$^{18}$O isotope labeling experiments: Both Au$_{SA}$-MnFeCoNiCu LDH and MnFeCoNiCu LDH were firstly activated by electrochemical CV method (0–0.8 V vs. Hg/HgO, 10 CV cycles) in $^{18}$O-labeled KOH electrolyte, and then carried out OER test (10 mA cm$^{-2}$ for 30 min) in 1.0 M KOH with regular H$_2$O. Subsequently, the gas was collected for GC-MS analysis.

Mass activity calculation:

$$Mass\ activity = j/m \tag{3}$$

Here j is the current density at a certain voltage and m is the catalyst loading.

DFT details. Using the Vienna ab initio (VASP) software, all DFT calculations were carried out by the projector-augmented wave method, the Perdew-Burke-Ernzerhof and the generalized gradient approximation were employed as the functional form and the description of the electron exchange and associated energies, respectively. For calculation parameters, the cut-off energy was set to 450 eV, and the convergence criteria for force and energy were set at 0.01 eV Å$^{-1}$ and $10^{-5}$ eV, respectively. Using the Monkhorst-Pack scheme of ($3 \times 3 \times 1$), sampling of the Brillouin zone was carried out for all model optimizations, and all plates were added with a 15 Å vacuum layer to separate their periodicity. For all calculations of density of states, the K point of the Brillouin zone is taken as ($9 \times 9 \times 1$). The Hubbard-U correction (DFT + U method) was applied to improve the description of localized metal d-electrons in the Au$_{SA}$-MnFeCoNiCu LDH and MnFeCoNiCu LDH systems. the value of U was set as 3.9, 4.0, 3.3, 6.0 and 3.87 eV for Mn, Fe, Co, Ni and Cu, respectively.

## Reporting summary

Further information on research design is available in the Nature Portfolio Reporting Summary linked to this article.

## Data availability

The data supporting the findings of this study are available within the article and Supplementary Information. All other relevant source data are available from the corresponding authors upon reasonable request.

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

## Acknowledgements

This work was supported by the Natural Science Foundation of Hebei Province (E2022202035, H.L., F.W., L.L.) and Natural Science Foundation of Hebei Province (B2021202019, Y.L.). P.Z., Y.Z, W.P., C.C. and S.Z. was unfunded. Wenhao Yuan is thanked for the contributions to figure typography and visual appeal.

## Author contributions

H.L. conceived the research idea. F.W. designed the experiments, prepared the materials, performed most of the characterizations and conducted the theoretical calculations. Y.Z. assisted in the characterizations. W.P. provide the synchrotron radiation source. Y.L. assisted in the DFT + U calculations. L.L. provided the Raman instrument and C.C. provided the theoretical computational resources. F.W. and H.L. drafted the manuscript, and S.Z. and P.Z. contributed to extensive revisions. All the co-authors contributed to the discussion and commented on the manuscript.

## Competing interests

The authors declare no competing interests.
