## [Peer Review File · Nature Communications]

Activating Lattice Oxygen in High-Entropy LDH for Robust and Durable Water OxidationREVIEWER COMMENTS

Reviewer #1 (Remarks to the Author):

The authors reported a high-entropy MnFeCoNiCu layered double hydroxide decorated with Au single atoms and O vacancies for the oxygen evolution reaction in alkaline media. While this topic is interesting and important, major revisions need to be done before it can be accepted for publication, especially in the DFT part.

1. There are similar works on high-entropy LDH (MnFeCoNiCr) for OER, and exhibiting even better performance ($\eta=218$ mV at 50 mA cm⁻², 400 mA cm⁻² for 600 h) published in Chemical Engineering Journal 466 (2023) 143352; Single-atom Au on NiFe LDH published in J. Am. Chem. Soc. 2018, 140, 3876–3879. Please carefully articulate the differences between the current work and these published works.
2. In Figure 3a, the authors need to indicate whether the LSV curves are IR corrected or not, for the Au-MnFeCoNiCu LDH curve it is more like over-corrected because the overpotential is even smaller at higher current densities above 200 mA cm⁻². (similar case in Figure 4a-b)
3. There are big problems in the DFT section, first of all, the authors did not apply Hubbard U correction (DFT+U) for the strongly correlated 3d electrons of MnFeCoNiCu, and therefore all the data are invalid in this sense.
4. The figures in SI are blurry and need to be replaced with higher quality figs. Half of the hydrogen termination of the LDH models in Figure S27-28, S31-34 is missing, the author need to check whether the OER processes are correctly calculated with right models.
5. Each elementary OER steps in the free energy diagrams in Fig. 5e and SI need to be labelled clearly. The calculated overpotentials in Fig. 5e are too large to be comparable to the experimental values, and again, DFT+U calculations with correct active site structures need to be re-investigated.
6. The OER is a four-electron transfer process, one of the elementary steps in LOM should not involve electron transfer, i.e., the O₂ desorption step, so the free energy diagrams at applied potential of 1.23 V need to be re-plotted even after the DFT+U calculations.
7. More importantly, how entropy affects the OER process needs to be reflected in theoretical studies, that is, to elucidate the fundamental necessity of high entropy.

Reviewer #2 (Remarks to the Author):

The authors fabricated a novel high-entropy MnFeCoNiCu layered double hydroxide decorated with Au single atoms and O vacancies (AuSA-MnFeCoNiCu LDH). A number of points need clarifying and certain statements require further justification. I think that the present manuscript is not adequate for the publication in this journal. There are given below:

- (1) Is it pristine LSV curves In Fig. 3a. I feel that the curves of Fig. 3a is similar with the Fig. S16, which indicate that the data is not a real measured curve.
- (2) The morphology of the aged sample in Fig. 3g should be provided.
- (3) The LSVs of the samples with different compositions should be provided for the performance comparison. How to determine the optimal composition for the OER in an alkaline medium?
- (4) The Mn, Fe and Co species of as-prepared sample have not a big influence on the OER performance. What is meaning of these elements existed in the as-prepared sample.
- (5) A novel high-entropy MnFeCoNiCu layered double hydroxide decorated with Au single atoms and O vacancies (AuSA-MnFeCoNiCu LDH) displays a low overpotential of 213 mV at 10 mA cm⁻² and exceptional stability with 700 hours of continuous operation at ~ 100 mA cm⁻². I think that the as-prepared sample is not very good performance in comparison with the current advanced electrocatalysts. Furthermore, the Au incorporation greatly increases the cost, leading to a difficulty in practical application.
- (6) The LOM usually results in a decrease of the OER stability owing to a easy structural collapse. Hence, the manuscript has a contradiction on the LOM and OER performance.

Reviewer #3 (Remarks to the Author):

Comments to authors

This manuscript reported a novel high-entropy MnFeCoNiCu layered double hydroxide (LDH) decorated with Au single atom and O vacancies. DFT calculations and various spectroscopic methods were used to validate the transformation of the OER mechanism from AEM to LOM, and the weakened metal-oxygen bonds during the water oxidation upon Au incorporation. Although the OER activity of the obtained high entropy-LDH is not impressive as compared to the other nanostructured high entropy alloy-based electrocatalysts, this work provided a method to prepare Au-incorporated high entropy-LDH for water splitting. In addition, the authors proposed a new mechanism for OER and validated it by DFT calculation. Therefore, I believe that this work can be accepted after substantial revisions and addressing the following comments.

1. Why did the authors choose Au instead of other metals? What is the difference between Au incorporation and the incorporation of other metal atoms? I recommend that authors should provide the justification.
2. In Fig. S1, the elemental mapping is not clear to identify the uniform metal distribution. Similarly, Fig. S4 is not clear to see the reported BE shift. So, authors should provide high-resolution images in the supporting information.
3. Is it a high-entropy material? How did the authors confirm it? There are many factors that define a high-entropy material. I suggested that authors should provide evidence for their claim.
4. How did the authors confirm the 3-layer formation of LDH from the AFM image in Fig. 1c? It is suggested to provide a more detailed explanation.
5. What is the lattice d-spacing value after Au incorporation? Is there any changes? I recommended authors to take the HRTEM image for the MnFeCoNiCu-LDH.
6. What is the catalyst loading? How did the authors calculate the mass activity?
7. From Fig. 3e, why did both materials in the case $R\eta/j$ exhibit the opposite trend? It is not clear, the authors should provide more clear explanation in this section.
8. What is the solution resistance? What is the iR corrections percentage?
9. The AuSA-MnFeCoNiCu-LDH showed considerably lower Tafel slope than commercial RuO₂ and IrO₂? What is the Tafel slope of IrO₂? Does the lower Tafel slope indicate a change in the reaction mechanism?
10. A few recent works of literature that are relevant to this work need to be cited in the "Introduction" and "Results" sections, such as "J. Mater. Chem. A, 2021, 9, 16841-16851 and J. Mater. Chem. A, 2020, 8, 11938-11947".
11. How about the HER activity of AuSA-MnFeCoNiCu-LDH? I suggested that the authors show it in the supporting information.

Reviewer #1 (Remarks to the Author):

The authors reported a high-entropy MnFeCoNiCu layered double hydroxide decorated with Au single atoms and O vacancies for the oxygen evolution reaction in alkaline media. While this topic is interesting and important, major revisions need to be done before it can be accepted for publication, especially in the DFT part.

1. There are similar works on high-entropy LDH (MnFeCoNiCr) for OER, and exhibiting even better performance ($\eta=218$ mV at 50 mA cm^{-2} , 400 mA cm^{-2} for 600 h) published in *Chemical Engineering Journal* 466 (2023) 143352; Single-atom Au on NiFe LDH published in *J. Am. Chem. Soc.* 2018, 140, 3876–3879. Please carefully articulate the differences between the current work and these published works.

Answer: Thank you very much for this valuable question. Apparently, the mentioned works (*Chem. Eng. J.*, 2023, 466, 143352; *J. Am. Chem. Soc.* 2018, 140, 3876) are similar with our work in terms of either high-entropy materials system or single-atom regulating strategy. However, there exists essential differences between the current work and these published works, which are described as follows.

(1) Unlike the work on high-entropy (FeNiCoMnCr) LDH published in *Chem. Eng. J.* (2023, 466, 143352) that emphasizes the binder-free high-entropy LDH structure and the high activity and stability towards OER, our work is mainly focusing on the successful regulation of the OER mechanism of high-entropy (MnFeCoNiCu) LDH from traditional AEM to LOM, and gaining a deep insight into the active mechanism on the lattice oxygen in high-entropy LDH. The high-entropy LDH based on LOM is firstly reported by this work, and exhibits the state-of-the-art performance especially in terms of stability among the OER catalysts based on LOM (*Nat Energy*, 2019, 4, 329–338; *Nat Commun.* 2020, 11, 4066; *Nat. Commun.* 2022, 13, 2191). Sure, we agree that the performance of the catalyst in our work is slightly inferior to the work published in *Chem. Eng. J.* (2023, 466, 143352). However, given that LOM can break the theoretical limit of AEM, we have a faith that designing novel high-entropy materials based on LOM will provide a new way to explore the robust and durable

electrocatalysts towards OER.

(2) For the work about single-atom Au on NiFe LDH (^sAu/NiFe LDH) published in *J. Am. Chem. Soc.* (2018, 140, 3876–3879), the incorporation of single-atom Au is aimed to optimize the charge density of active Fe site in LDH through the electronic coupling at the Au-catalyst interface. However, the aforementioned work doesn't involve the LOM mechanism at all, which is very different from our work that the introduction of single-atom Au is designed for activating the lattice oxygen in high-entropy LDH and triggering LOM. Additionally, the catalytic activity and stability in this work ($\eta=213$ mV at 50 mA cm^{-2} , $\sim 100 \text{ mA cm}^{-2}$ for 700 h) is superior to the work published in *J. Am. Chem. Soc.* 2018, 140, 3876–3879 ($\eta=237$ mV at 10 mA cm^{-2} , $\sim 100 \text{ mA cm}^{-2}$ for 20 h).

Finally, to highlight the novelty and significance of our work, we have improved the statement in the introduction section, emphasizing the novelty and significance of the transformation of the oxygen evolution mechanism in high-entropy LDH catalysts through the incorporation of single-atom Au. The detailed description is shown in the revised introduction section (Page 3, paragraph 1; Page 4, paragraph 1), which is marked with red color.

2. In Figure 3a, the authors need to indicate whether the LSV curves are IR corrected or not, for the Au-MnFeCoNiCo LDH curve it is more like over-corrected because the overpotential is even smaller at higher current densities above 200 mA cm^{-2} . (similar case in Figure 4a-b)

Answer: All OER test curves in this work were 95% IR correction. To investigate the effect of different IR corrections on the OER activity of Au_{SA}-MnFeCoNiCu LDH, we carried out LSV tests at different IR compensations (95%, 80% and without compensation), and the results are shown in **Figure R1** and **Figure S15** in the revision. Under 80% and 95% IR compensation, both of the LSV curves show nearly vertical lines at high current densities over 100 mA cm^{-2} . Even more the LSV curve without IR compensation exhibit a fast reaction kinetics at high current density, suggesting that the excellent activities are not attributed to the IR compensation but the distinguished

intrinsic activity.

Additionally, it seems that the overpotential is even smaller at higher current densities above 200 mA cm^{-2} due to the fast reaction kinetics, but the accurate data shown in **Table R1** indicate that it's an appearance.

Figure R1. LSV curves of $\text{Au}_{\text{SA}}\text{-MnFeCoNiCu}$ LDH with different IR corrections.

Table R1. $\text{Au}_{\text{SA}}\text{-MnFeCoNiCu}$ LDH overpotential at different current densities.

Current density (mA cm^{-2})	10	100	200	300
Overpotential (mV)	213	260	262	264

Changes in the revised manuscript

Added discussion: We also carried out LSV tests for $\text{Au}_{\text{SA}}\text{-MnFeCoNiCu}$ LDH and MnFeCoNiCu LDH with different iR compensations (**Figure 3a** and **Figure S15**), which are discussed in **Supplementary note 1**.

3. There are big problems in the DFT section, first of all, the authors did not apply Hubbard U correction (DFT+U) for the strongly correlated 3d electrons of MnFeCoNiCu , and therefore all the data are invalid in this sense.

Answer: Thank you for your valuable suggestions. According to your recommendation, we have recalculated all the data by applying Hubbard U correction (DFT+U) due to the strongly correlated 3d electrons of MnFeCoNiCu . The calculated results are updated,

as shown in **Figure 5**, **Figures S40-S51** and **Table S6-S9**. Although the updated data with DFT+U differ from those without Hubbard U correction, the conclusions are consistent. The DFT+U calculations further confirm that the introduced Au single atoms and the oxygen vacancies can synergistically upshift O 2p orbitals and weaken metal-oxygen bonds to activate the lattice oxygen and lower the energy barrier of LOM, which facilitates the lattice oxygen oxidation and accelerates OER kinetics.

4. The figures in SI are blurry and need to be replaced with higher quality figs. Half of the hydrogen termination of the LDH models in Figure S27-28, S31-34 is missing, the author need to check whether the OER processes are correctly calculated with right models.

Answer: Thanks for your suggestion. (1) We have improved the quality of the figures in SI. Please see the revision. (2) In this work, we choose the high-entropy oxyhydroxides ($\text{Au}_{\text{SA}}\text{-MnFeCoNiCuOOH}$) rather than pristine LDH ($\text{Au}_{\text{SA}}\text{-MnFeCoNiCu LDH}$) as calculated model, which is due to the surface reconfiguration of LDH phase during the OER process. The results of in situ Raman spectroscopy (**Figure S26**) and EDS-mapping (**Figure S27**) characterization jointly demonstrate the formation of high-entropy $\text{Au}_{\text{SA}}\text{-MnFeCoNiCuOOH}$ phase on the surface of high-entropy LDH during OER, all of which results are consistent with the published works on LDH (*Nano Energy*, 2018, 44, 181-190; *ACS Catal.* 2023, 13, 7, 4799–4810; *small* 2301610), so we consider high-entropy oxyhydroxides instead of LDH as the real active species. Given that oxyhydroxides only have half of the hydrogen termination of the LDH models, the construction of calculation model in the original manuscript is appropriate.

Figure S26. In situ Raman spectra of Au_SA-MnFeCoNiCu LDH. With the increase of the OER potential, two narrow peaks corresponding to the Ni^{II}-O bond gradually weaken until they disappear, while a broad peak gradually appears, which corresponds to the NiOOH species. It indicates that with the OER process, the Ni(OH)_x on the Au_SA-MnFeCoNiCu LDH surface gradually transforms to the non-characterized NiOOH.

Figure S27. Element mapping of Au_SA-MnFeCoNiCu LDH after 50 h long-term OER test.

5. Each elementary OER steps in the free energy diagrams in Fig. 5e and SI need to be labelled clearly. The calculated overpotentials in Fig. 5e are too large to be comparable

to the experimental values, and again, DFT+U calculations with correct active site structures need to be re-investigated.

Answer: Thanks for your suggestion. We have clearly labelled each elementary OER steps in the free energy diagrams in **Figure 5e** and SI. Moreover, we have recalculated all the data with Hubbard U correction (DFT+U) for the strongly correlated 3d electrons of MnFeCoNiCu. According to the literatures (*Nat. Chem.*, 2017, 9, 457-465; *Nat. Catal.*, 2018, 1, 339- 348; *ACS Catal.* 2023, 13, 11, 7698-7706; *Nat. Commun.*, 2022, 13, 2191), the values of U were set to 3.9, 4.0, 3.3, 6.0 and 3.87 eV for Mn, Fe, Co, Ni, Cu, respectively. Compared with the original result without DFT+U (1.14 eV), the calculated overpotential with DFT+U (0.81 eV) is more in line with the experimental result. Certainly, the calculated overpotential with DFT+U is still larger than the experimental value, which may be attributed to the solvent effect and the electric field effect caused by the double electric layer at the interface are not considered (*J. Phys. Chem. C*, 2020, 124, 14581–14591; *Nat. Commun.*, 2022, 13, 174). Even so, the calculated results with DFT+U is still reliable, because it can reflect the trends among samples, which is consistent with many published works (*Nat. Commun.*, 2020, 11, 4066; *Angew. Chem. Int. Ed.* 2021, 60 (52), 27126-27134.).

6. The OER is a four-electron transfer process, one of the elementary steps in LOM should not involve electron transfer, i.e., the O₂ desorption step, so the free energy diagrams at applied potential of 1.23 V need to be re-plotted even after the DFT+U calculations.

Answer: Thanks for your reminder, we have re-plotted all free energy diagrams concerning the LOM after the DFT+U calculations, removing the O₂ desorption step that does not involve electron transfer. Please see **Figure 5e** and **Figure S48**.

Figure 5e. Computed free energies (ΔG) of OER steps on $\text{Au}_{\text{SA}}\text{-MnFeCoNiCuOOH}$ and MnFeCoNiCuOOH .

Figure S48. Free energies of OER via LOM on MnFeCoNiCuOOH with only Au atom and MnFeCoNiCuOOH with only O vacancy.

7. More importantly, how entropy affects the OER process needs to be reflected in theoretical studies, that is, to elucidate the fundamental necessity of high entropy.

Answer: In order to elucidate the effect of high entropy on the OER process from the viewpoint of theoretical studies, we constructed a high-entropy $\text{Au}_{\text{SA}}\text{-MnFeCoNiCuOOH}$ model with disordered 3d transition metals and a non-high-entropy $\text{Au}_{\text{SA}}\text{-MnFeCoNiCuOOH}$ model with ordered 3d transition metals, as illustrated in

Figure S49. We calculate the formation energies of the high-entropy and non-high-entropy samples, respectively, and the results show that the formation energy of the high-entropy sample (-1.75 eV) is significantly lower than that of the non-high-entropy sample (-0.21 eV), suggesting that the high-entropy effect can strengthen the structure stability of the catalyst (**Figure S50**). Moreover, given that the easy leaching of metal species next to oxygen vacancies in LOM-based electrocatalysts during OER (*Nat. Energy*, 2017, 2, 16189; *Joule*, 2021, 5(8):2164-2176), we also calculated the binding energies of Ni species next to oxygen vacancies in high-entropy and non-high-entropy samples, which indicates that the binding energy of Ni next to oxygen vacancies in high-entropy sample (-1.35 eV) is more negative than that of non-high-entropy sample (-0.25 eV), implying that high-entropy effect can inhibit the leaching of metal species and boost the structure stability during lattice-oxygen oxidation process, which is responsible for the excellent OER stability (**Figure S51**). Relevant content has been added to the revised manuscript.

Figure S49. high-entropy $\text{Au}_{\text{SA}}\text{-MnFeCoNiCuOOH}$ model with disordered 3d transition metals and non-high-entropy $\text{Au}_{\text{SA}}\text{-MnFeCoNiCuOOH}$ model with ordered 3d transition metals (Gray: Ni, gold: Au, orange: Co, navy: Cu, purple: Fe, dark red: Mn, red: O, and white: H).

Figure S50. The formation energy of high-entropy structure and non-high-entropy structure.

Figure S51. The binding energy of Ni next to O vacancy in high-entropy structure and non-high-entropy structure.

Reviewer #2 (Remarks to the Author):

The authors fabricated a novel high-entropy MnFeCoNiCu layered double hydroxide decorated with Au single atoms and O vacancies (Au_{SA}-MnFeCoNiCu LDH). A number of points need clarifying and certain statements require further justification. I think that the present manuscript is not adequate for the publication in this journal. There are given below:

1. Is it pristine LSV curves In Fig. 3a. I feel that the curves of Fig. 3a is similar with the Fig. S16, which indicate that the data is not a real measured curve.

Answer: Thank you for your question. The LSV curves of **Figure 3a** are pristine curves. The similarity between **Figure 3a** and **Figure S16** (**Figure S22** in the revision) is because the LSV curves of **Figure S16** are ECSA-normalized LSV curves of **Figure 3a**, which are obtained by the current densities of LSV curves in **Figure 3a** divided by ECSA value ($\text{Current density}^{\text{ECSA}} = \text{Current density} / \text{ECSA}$, $\text{ECSA} = C_{\text{dl}} / C_s$, with the specific details in the “Methods” section). In order to clarify the difference between **Figure 3a** and **Figure S16**, we merge **Figure 3a** and **Figure S16** into one graph (**Figure R2**). Although the LSV curves and the ECSA-normalized LSV curves are similar, they are different, especially for MnFeCoNiCu LDH. As for Au_{SA}-MnFeCoNiCu LDH, the high similarity between the LSV curve and the ECSA-normalized LSV curve stems from the sharp increase in current densities of LSV curve, resulting in the similar sharp increase in ECSA-normalized current densities, which makes them look similar. However, the enlarged figure in **Figure R2** shows that they are different.

Figure R2. Comparison of LSV and ECSA normalized LSV.

2. The morphology of the aged sample in Fig. 3g should be provided.

Answer: Thank you for your suggestion. According to your suggestion, we have added the SEM and TEM images of the samples after 700 h stability test (**Figure S29**) in the revised supporting information. The results show that the aged sample maintains its nanosheet morphology after long-term stability testing. Moreover, the HRTEM image (**Figure S30**) shows that Au_{SA}-MnFeCoNiCu LDH after 700 h stability test exhibits an amorphous layer with approximately 10 nm, which is similar with the sample after initial 50 h stability test (**Figure S25**), implying the maintenance of the structure stability, which accounts for the excellent stability.

Figure S29. (a) SEM and (b) TEM images of Au_{SA}-MnFeCoNiCu LDH after 700 h long-term OER test.

Figure S30. HRTEM image of Au_{SA}-MnFeCoNiCu LDH after 700 h long-term OER test.

Figure S25. HR-TEM images of Au_{SA}-MnFeCoNiCu LDH after 50 h long-term OER test.

3. The LSVs of the samples with different compositions should be provided for the performance comparison. How to determine the optimal composition for the OER in an alkaline medium?

Answer: Thank you for your advice. Based on your suggestion, we have synthesized 20 high-entropy LDHs materials decorated with single-atom Au with different compositions (Au_{SA}-HE LDHs) besides Au_{SA}-MnFeCoNiCu LDH. The XRD, EDS and elemental mapping image as shown in **Figure S16**, **Figure S17** and **Table S3** confirm the successful formation of single high-entropy phase. The LSV curves of 20 high-entropy samples with different composition are provided in **Figure S18**. Moreover, the LSV curves after 10000 CV cycles of the samples are also offered to reflect the stability (**Figure S19** and **Figure S20**). It can be clearly seen that Au_{SA}-MnFeCoNiCu LDH exhibit the lowest overpotential at 10 mA cm⁻² (213 mV) and the smallest positive shift (7 mV) after 10000 CV cycles among all the samples, implying the best OER activity and stability, which also indicate the optimal composition.

Figure S16. XRD spectra of 20 Au_{SA}-HE LDHs.

Figure S17. Elemental mapping of 20 Au_{SA}-HE LDHs, scale bar is 250 nm.

Table S3. EDS results of 20 Au_{SA}-HE LDHs.

	Cr (at%)	Mn (at%)	Fe (at%)	Co (at%)	Ni (at%)	Cu (at%)	Zn (at%)	O (at%)	Au (at%)
Au _{SA} - CrMnFeCoNi LDH	3.65	4.68	5.91	3.68	5.03	-	-	76.84	0.21
Au _{SA} - CrMnFeCoCu LDH	3.64	4.37	4.38	3.88	-	6.81	-	76.66	0.26
Au _{SA} - CrMnFeCoZn LDH	4.51	3.46	4.43	4.45	-	-	5.33	77.52	0.3
Au _{SA} - CrMnFeNiCu LDH	3.75	4.79	5.86	-	6.14	5.07	-	74.14	0.25
Au _{SA} - CrMnFeNiZn LDH	5.41	4.57	5.21	-	4.18	-	5.58	74.77	0.28
Au _{SA} - CrMnFeCuZn LDH	4.31	4.47	5.91	-	-	5.74	4.91	74.37	0.29
Au _{SA} - CrMnCoNiCu LDH	3.32	3.64	--	4.79	5.41	4.63	-	78.00	0.21
Au _{SA} - CrMnCoNiZn LDH	4.83	6.73	-	5.26	5.84	-	4.74	72.38	0.22
Au _{SA} - CrMnCoCuZn LDH	4.72	4.26	-	4.42	-	4.75	4.45	77.20	0.20
Au _{SA} - CrMnNiCuZn LDH	4.31	4.32	-	-	5.39	4.36	5.56	75.81	0.25
Au _{SA} - CrFeCoNiCu LDH	5.36	-	4.38	5.63	5.83	4.92	-	73.64	0.24
Au _{SA} - CrFeCoNiZn LDH	4.18	-	4.80	4.78	5.50	-	5.55	74.95	0.24
Au _{SA} - CrFeCoCuZn LDH	4.35	-	4.51	5.00	-	4.78	5.83	75.26	0.27
Au _{SA} - CrFeNiCuZn LDH	5.38	-	4.94	-	5.35	4.75	4.59	74.78	0.21
Au _{SA} - CrCoNiCuZn LDH	5.18	-	-	5.82	4.94	5.86	5.97	72.00	0.23
Au _{SA} - MnFeCoNiZn LDH	-	3.94	4.13	3.84	4.14	-	4.38	79.32	0.25

Au _{SA} -MnFeCoCuZn LDH	-	4.73	4.76	3.43	-	3.49	3.70	79.63	0.26
Au _{SA} -MnFeNiCuZn LDH	-	5.67	4.30	-	5.03	5.24	5.18	74.30	0.28
Au _{SA} -MnCoNiCuZn LDH	-	5.42	-	4.53	5.53	4.46	5.79	74.02	0.25
Au _{SA} -FeCoNiCuZn LDH	-	-	4.74	5.63	4.45	5.94	4.91	74.04	0.29

Figure S18. LSV curves of 20 Au_{SA}-HE LDHs.

Figure S19. LSV shift of $\text{Au}_{\text{SA}}\text{-MnFeCoNiCu LDH}$.

Figure S20. LSV shift of comparative 20 $\text{Au}_{\text{SA}}\text{-HE LDHs}$.

Changes in the revised manuscript

Added discussion: Besides Au_{SA}-MnFeCoNiCu LDH, we also successfully synthesized 20 Au-decorated high-entropy LDHs materials (Au_{SA}-HE LDHs) with different quinary transition-metals composition (Cr, Mn, Fe, Co, Ni Cu or Zn), and tested their OER performance (**Figure S16-S20 and Table S3**), details as shown in **Supplementary note 2**. Considering the low η and the small Tafel slope, the as-synthesized Au_{SA}-MnFeCoNiCu LDH exhibits better OER activity than other Au_{SA}-HE LDHs in this works (**Figure S18**) and most reported high-entropy electrocatalysts (**Figure 3d**).

4. The Mn, Fe and Co species of as-prepared sample have not a big influence on the OER performance. What is meaning of these elements existed in the as-prepared sample. Answer: In order to clarify the role of Mn, Fe and Co species of as-prepared samples on the OER performance, we have conducted controlled experiments. First, we have successfully prepared quaternary Mn-free Au_{SA}-FeCoNiCu LDH, Fe-free Au_{SA}-MnCoNiCu LDH and Co-free Au_{SA}-MnFeNiCu LDH, respectively, as shown in **Figure R3, Figure R4 and Table R2**. Then, the LSV curves of the three samples before and after the 10,000 CV cycles are illustrated in **Figure R5**. It can be clearly observed that the OER activities of all quaternary samples (Au_{SA}-FeCoNiCu LDH: 319 mV@10 mA cm⁻²; Au_{SA}-MnCoNiCu LDH: 354 mV@10 mA cm⁻²; Au_{SA}-MnFeNiCu LDH: 267 mV@10 mA cm⁻²) are inferior to quinary Au_{SA}-MnFeCoNiCu LDH (213 mV@10 mA cm⁻²). Furthermore, the stability of all quaternary samples decreased rapidly relative to Au_{SA}-MnFeCoNiCu LDH (**Figure S19**), regardless of which element was removed. Hence, we speculate that the absence of these elements will lead to low configurational entropy compared to quinary Au_{SA}-MnFeCoNiCu LDH, and low configurational entropy may weaken entropy stabilization effect that plays a key role in the stability of the electrocatalysts. The impact of the entropy effect on the stability of electrocatalysts in theoretical studies has been discussed in detail. Please refer to the reply of the 6th question.

Figure R3. XRD spectra of 3 quadruple Au_{SA}-LDHs.

Figure R4. Elemental mapping of 3 quadruple Au_{SA}-LDHs.

Table R2. EDS results of 3 quadruple Au_{SA}-LDHs.

	Mn (at%)	Fe (at%)	Co (at%)	Ni (at%)	Cu (at%)	O (at%)	Au (at%)
Au _{SA} -FeCoNiCu LDH	-	6.23	4.39	5.58	7.88	75.69	0.23
Au _{SA} -MnCoNiCu LDH	5.87	-	4.91	5.00	6.75	77.22	0.25
Au _{SA} -MnFeNiCu LDH	8.16	6.58		8.59	5.12	71.34	0.21

Figure R5. LSV shift of 3 quadruple Au_{SA}-LDHs.

Figure S19. LSV shift of Au_{SA}-MnFeCoNiCu LDH.

5. A novel high-entropy MnFeCoNiCu layered double hydroxide decorated with Au single atoms and O vacancies (Au_{SA}-MnFeCoNiCu LDH) displays a low overpotential of 213 mV at 10 mA cm⁻² and exceptional stability with 700 hours of continuous operation at ~100 mA cm⁻². I think that the as-prepared sample is not very good performance in comparison with the current advanced electrocatalysts. Furthermore, the Au incorporation greatly increases the cost, leading to a difficulty in practical application.

Answer: (1) In fact, from the view of the catalytic activity, our as-prepared sample is not the best among the current advanced electrocatalysts, but it's superior to commercial IrO₂ and most of OER electrocatalysts reported recently (*Nat Commun.* 2022, 13, 2191; *Nat Energy*, 2019, 4, 329–338; *Nat Commun.*, 2022, 13, 2191). Most importantly, the highlight in this work is realizing the successful regulation of the OER mechanism of high-entropy LDH from traditional AEM to LOM, and gaining a deep insight into the

active mechanism on the lattice oxygen in high-entropy LDH. The high-entropy LDH based on LOM is firstly reported, and exhibits the good performance especially in terms of stability among the OER electrocatalysts based on LOM (*Nat Commun.*, 2022, 13, 2473; *Angew. Chem. Int. Ed.* 2021, 60, 27126-27134; *Nat Commun.*, 2021, 12, 3992). Given that LOM can break the theoretical limit of AEM, we firmly believe that designing novel high-entropy materials based on LOM will provide a new way to explore the robust and durable electrocatalysts towards OER.

(2) Although Au is precious and expensive, its usage in this work is rarely small (only 1.1 wt%) because Au is incorporated into the sample in the form of superficial single atoms. Recently, a large number of noble-metal single-atom electrocatalysts towards OER and beyond including Ir, Au and Ru single atoms et al. have been reported (*Nat Commun.*, 2022, 13, 7754; *J. Am. Chem. Soc.* 2021, 143, 13605–13615; *Nat. Catal.*, 2021, 4, 1012-1023), implying the potentially practical application due to its ultra-low loading and high performance.

Finally, to highlight the significance of this work, we have improved the introduction section, emphasizing the novelty and significance of the regulation of the oxygen evolution mechanism in high-entropy LDH catalysts through the incorporation of single-atom Au. The detailed description is shown in the revised introduction section (Page 3, paragraph 1; Page 4, paragraph 1), which is marked with red color.

6. The LOM usually results in a decrease of the OER stability owing to a easy structural collapse. Hence, the manuscript has a contradiction on the LOM and OER performance. Answer: Most of the electrocatalysts based on LOM previously reported (*Nat. Commun.*, 2022, 13, 2191; *Nat. Energy*, 2019 4, 329–338; *Nat. Commun.*, 2021, 12, 3992) suffer from poor stability, which becomes a great challenge in this fields, despite that LOM can break the theoretical limit of AEM. The motivation of this study is aimed to address this challenge, and **we successfully resolve this contradiction by utilizing high-entropy effect**. The following description is how we solve this challenge.

Recent studies reveal that the poor stability of LOM-based electrocatalysts is

attributed to the structure collapse, which may be attributed to the following reason: the metal species next to oxygen vacancies generated by lattice oxygen oxidation during OER are easy to leach from the lattice due to unsaturated oxygen coordination, leading to the structure collapse (*Joule*, 2021, 5(8):2164-2176); Inspired by the unique entropy-stabilized effect and sluggish diffusion effect of high-entropy materials, we speculate that the high-entropy OER electrocatalysts based on LOM may be a kind of promising candidate to solve the stability issue by inhabiting metal species leaching. Fortunately, the results agree well with our assumption. Based on ICP-OES test results, the leaching of metal ions in high-entropy Au_{SA}-MnFeCoNiCu LDH is successfully suppressed, as shown in **Figure S33**. The as-prepared high-entropy Au_{SA}-MnFeCoNiCu LDH based on LOM exhibits 700 h long-term durability at ~100 mA cm⁻², which is the most advanced LOM-based OER electrocatalyst in terms of stability, to our best knowledge. (*Nat. Energy*, 2019, 4, 329–338; *Nat. Commun.*, 2021, 12, 3992; *Adv. Mater.*, 2022, 34(50): 2107956)

To gain a deep insight into the impact of entropy effect on the stability, we also conduct DFT calculation. A high-entropy Au_{SA}-MnFeCoNiCuOOH model with disordered 3d transition metals and a non-high-entropy Au_{SA}-MnFeCoNiCuOOH model with ordered transition metals are built, as shown in **Figure S49**. The calculated results (**Figure S50 and Figure S51**) validate that the high-entropy sample not only exhibits a better structure stability than non-high-entropy sample, but also inhabits the leaching of metal species.

In summary, combining the experimental results and the theoretical studies, we successfully dissolve the contradiction between LOM and OER stability through utilizing high entropy effects, providing a new approach to explore the robust and durable LOM-based electrocatalysts towards OER.

Figure S33. After the stability test of NiFe LDH and Au_{SA}-MnFeCoNiCu LDH, the proportion of Fe element dissolved in the electrolyte to the Fe element in the original sample, which stands for the Fe dissolution percentage.

Figure S49. high-entropy Au_{SA}-MnFeCoNiCuOOH model with disordered 3d transition metals and non-high-entropy Au_{SA}-MnFeCoNiCuOOH model with ordered 3d transition metals (Gray: Ni, gold: Au, orange: Co, navy: Cu, purple: Fe, dark red: Mn, red: O, and white: H).

Figure S50. The formation energy of high-entropy structure and non-high-entropy structure.

Figure S51. The binding energy of Ni next to O vacancy in high-entropy structure and non-high-entropy structure.

Reviewer #3 (Remarks to the Author):

Comments to authors

This manuscript reported a novel high-entropy MnFeCoNiCu layered double hydroxide (LDH) decorated with Au single atom and O vacancies. DFT calculations and various spectroscopic methods were used to validate the transformation of the OER mechanism from AEM to LOM, and the weakened metal-oxygen bonds during the water oxidation upon Au incorporation. Although the OER activity of the obtained high entropy-LDH is not impressive as compared to the other nanostructured high entropy alloy-based electrocatalysts, this work provided a method to prepare Au-incorporated high entropy-LDH for water splitting. In addition, the authors proposed a new mechanism for OER and validated it by DFT calculation. Therefore, I believe that this work can be accepted after substantial revisions and addressing the following comments.

1. Why did the authors choose Au instead of other metals? What is the difference between Au incorporation and the incorporation of other metal atoms? I recommend that authors should provide the justification.

Answer: Thank you for your recommendation. We choose Au instead of other metals

based on the following considerations: (1) First, the chosen Au in this work is to activate the lattice oxygen in high-entropy LDH. Previous studies validate that increasing the covalency of metal-oxygen bonds is critical to triggering lattice-oxygen oxidation (*Nat. Chem.*, 2017, 9, 457–465). Given that Au has the greatest electronegativity (2.54) of all metals, we speculate the incorporation of Au atoms into high-entropy LDH may increase metal-oxygen covalency, triggering LOM in high-entropy LDH. (2) Second, Au is a kind of inert metal element that has strong alkali resistance and electrochemical stability (*Nat. Commun.*, 2021, 12, 1675; *J. Am. Chem. Soc.*, 2018, 140, 3876–3879; *Nano Today*, 2007, 2(4):14-18), possibly avoiding the leaching during anodic oxidation, which is beneficial for the structure stability of the catalyst. Therefore, we choose Au instead of other metals as the incorporated atoms, and the results agrees well with our assumption. The related description has been added in the revised “introduction” section in order to let the readers understand the motivation of this work.

To further reveal the relationship between the electronegativity of the incorporated metal atoms and the oxygen evolution mechanism, we choose other three metals with different electronegativity (Ru: 2.2, Pt: 2.3 and Ag: 1.9) for comparison. The structure characterization results validate the successful incorporation of single-atom Ru, Pt and Ag (**Figure S35-S37**) into high-entropy MnFeCoNiCu LDH. The LSV curve of different single-atom incorporation on MnFeCoNiCu LDH is displayed in **Figure S38**. Subsequently, we investigate the pH-dependent properties of high-entropy LDH catalysts incorporated with different single atoms, as shown in **Figure S39**, and the ρ^{RHE} value is utilized to evaluate the pH-dependent properties. The results reveal that there is a positive correlation between the electronegativity and the pH-dependent properties, implying that incorporating the single-atom metal with higher electronegativity is more conducive to triggering LOM. The relative discussion is added in the revised manuscript.

Figure S35. XRD patterns of different single-atom incorporation on MnFeCoNiCu LDH.

Figure S36. Element mapping of (a) Ru_{SA}-MnFeCoNiCu LDH, (b) Ag_{SA}-MnFeCoNiCu LDH and (c) Pt_{SA}-MnFeCoNiCu LDH.

Figure S37. AC-HAADF-STEM images of (a) Pt_{SA}-MnFeCoNiCu LDH, (b) Ru_{SA}-MnFeCoNiCu LDH and (c) Ag_{SA}-MnFeCoNiCu LDH.

Figure S38. LSV curves of Pt_{SA}-MnFeCoNiCu LDH, Ru_{SA}-MnFeCoNiCu LDH and Ag_{SA}-MnFeCoNiCu LDH.

Figure S39. ρ^{RHE} of different single-atom incorporation at 1.45 V vs. RHE.

Changes in the revised manuscript

Added discussion 1: Previous studies validate that increasing the covalency of metal-oxygen bonds is critical to triggering lattice-oxygen oxidation. Given that the covalency

of metal-oxygen bonds is determined by the electronegativity of metal, and Au has the greatest electronegativity (2.54) of all metals, we speculate the incorporation of Au atoms into high-entropy LDH may increase metal-oxygen covalency, triggering LOM. Also, Au is a kind of inert metal element that has strong alkali resistance and electrochemical stability, possibly avoiding the leaching during anodic oxidation, which is beneficial for the structure stability of the catalyst. Additionally, the introduction of a small amount of Au single atoms on the surface of high-entropy LDH instead of doping in the lattice with large amounts of Au atoms may be more realistic in consideration of the high cost of Au and the big difference in ion radius between Au and 3d transition elements.

Added discussion 2: Considering the greatest electronegativity of Au (2.54) among all transition metals, we conclude that the high electronegativity of Au may be responsible for the activation of the lattice oxygen in high-entropy LDH, because it may induce high metal-oxygen covalency (*Nat. Chem.*, 2017, 9, 457–465). To further reveal the relationship between the electronegativity of the incorporated metal atoms and the oxygen evolution mechanism, we choose other three metals with different electronegativity (Ru: 2.2, Pt: 2.3 and Ag: 1.9) for comparison, and the results validate that incorporating the single-atom metal with higher electronegativity is more conducive to triggering LOM in high-entropy LDH (**Figure S35-S39**), as detailed in **Supplementary note 4**.

2. In Fig. S1, the elemental mapping is not clear to identify the uniform metal distribution. Similarly, Fig. S4 is not clear to see the reported BE shift. So, authors should provide high-resolution images in the supporting information.

Answer: Thank you for your suggestion. We have improved the quality of **Figure S1** and **Figure S4** with higher resolution, which is shown in the revised manuscript. Now, the elemental mapping in **Figure S1** clearly shows the uniform distribution of elements, and the BE shift is also clearly indicated in **Figure S5** (Original **Figure S4**). Moreover, we also improved the image resolution of other figures in the supporting information.

Figure S1. (a) XRD patterns, (b) SEM image, (c) TEM image, (d) EDS spectrum and (e) mapping of MnFeCoNiCu LDH.

Figure S5. High-resolution XPS spectra of (a) Mn 2p, (b) Fe 2p, (c) Co 2p, (d) Ni 2p, (e) Cu 2p of Au_{SA}-MnFeCoNiCu LDH and MnFeCoNiCu LDH. (f) Au 4f of Au_{SA}-MnFeCoNiCu LDH.

3. Is it a high-entropy material? How did the authors confirm it? There are many factors that define a high-entropy material. I suggested that authors should provide evidence for their claim.

Answer: (1) We believe that the as-prepared Au_{SA}-MnFeCoNiCu LDH in this work is high-entropy materials. Despite the traditional range for high-entropy materials is limited to high-entropy alloys, this class of materials have been extended to high-entropy compounds including cationic and anionic high-entropy materials (*Nat. Rev. Mater.* 2020, 5, 295-309; *J. Energy Chem.*, 2021, 60, 121-126; *Adv. Mater.*, 2022, 34(6):2110511). In this work, our as-prepared Au_{SA}-MnFeCoNiCu LDH is cationic high-entropy compounds.

(2) The definition of a high entropy compounds is that single-phase solid solution consisting of five or more cations uniformly distributed in throughout the cationic lattice position in the crystal structure (*Nat. Rev. Mater.* 2020, 5, 295–309; *Nat Commun.*,2020, 11, 3908; *Angew. Chem. Int. Ed.*, 2021, 60(37):20253-20258;). This definition has four key points: 1. Five or more cations. 3. All cations uniformly distributed. 4. The crystalline structure is single phase. In this work, the as-prepared Au_{SA}-MnFeCoNiCu LDH has five cations (Mn, Fe, Co, Ni, Cu). In addition, elements mapping (**Figure 1h**) demonstrates a homogeneous distribution of all elements. Finally, XRD (**Figure S2**) demonstrates that the as-prepared Au_{SA}-MnFeCoNiCu LDH is a single phase with no other heterogeneous phases. Therefore, our as-prepared Au_{SA}-MnFeCoNiCu LDH is high entropy materials, which is also consistent with many published works (*Adv. Mater.*, 2022, 34 (26) 2110511; *J. Energy Chem.*, 2021, 60, 121-126).

Figure 1h. EDS elemental mapping of Mn, Fe, Co, Ni, Cu, O, Au. The scale bar is 1 μm .

Figure S2. XRD patterns of Au_{SA}-MnFeCoNiCu LDH.

4. How did the authors confirm the 3-layer formation of LDH from the AFM image in Fig. 1c? It is suggested to provide a more detailed explanation.

Answer: Recent studies have demonstrated that the thickness of single layer of high-entropy LDH is about 0.7~0.9 nm (*Adv. Mater.*, 10.1002/adma.202302860; *Nat Commun.*, 2014, 5, 4477; *Adv. Energy Mater.*, 2019, 9, 1900881;). Moreover, the layer spacing of LDH can be reflected by the diffraction peak of (003) plane in the XRD pattern (**Figure S2**), and the calculated value based on Bragg formula is 0.74 nm. Therefore, the as-prepared high-entropy LDH is comprised of 3 layers. The detailed explanation has been provided in the revised manuscript.

Changes in the revised manuscript

Original discussion: The atomic force microscope (AFM) results reveal that a typical Au_{SA}-MnFeCoNiCu LDH nanosheet was 3.4 nm thick and comprised three layers (**Figure 1c**).

Revised discussion: Atomic force microscopy (AFM) results show that a typical Au_{SA}-MnFeCoNiCu LDH nanosheet has a thickness of 3.4 nm (**Figure 1c**). Recent studies have demonstrated that the thickness of single layer of high-entropy LDH is about 0.7~0.9 nm (*Nat. Commun.*, 2014, 5, 4477; *Adv. Energy Mater.*, 2019, 9, 1900881; *Nano Res.*, 2018, 11, 195-205). Moreover, the layer spacing of LDH can be reflected by the diffraction peak of (003) plane in the XRD pattern (**Figure S2**), and the

calculated value based on Bragg formula is 0.74 nm. Therefore, the as-prepared Au_{SA}-MnFeCoNiCu LDH is about 3 layers.

5. What is the lattice d-spacing value after Au incorporation? Is there any changes? I recommended authors to take the HRTEM image for the MnFeCoNiCu-LDH.

Answer: Thanks for your suggestion. We have provided the lattice d-spacing value of high-entropy LDH before and after Au incorporation in the revision (**Figure S4** and **Figure S1d**). The lattice spacing of (101) plane in Au_{SA}-MnFeCoNiCu LDH is 0.261 nm, slightly larger than that in MnFeCoNiCu LDH (0.258 nm), which is attributed to that the atomic radius of Au single atoms is larger than that of 3d transition metal elements, resulting in the lattice expansion. The detailed description has been added in the revision.

Figure S4. HRTEM image of MnFeCoNiCu LDH.

Figure S1d. HRTEM image of Au_{SA}-MnFeCoNiCu LDH. The scale bar is 10 nm.

Changes in the revised manuscript

Original discussion: The clear lattice fringe with an interplanar distance of 0.261 nm shown in the high-resolution TEM (**Figure 1d**) corresponds to the (101) plane of Au_{SA}-MnFeCoNiCu LDH.

Revised discussion: The clear lattice fringe with an interplanar distance of 0.261 nm shown in the high-resolution TEM (**Figure 1d**) corresponds to the (101) plane of Au_{SA}-MnFeCoNiCu LDH. In contrast, the lattice spacing of MnFeCoNiCu LDH is 0.258 nm (**Figure S4**). The slight expansion of Au_{SA}-MnFeCoNiCu LDH in terms of interplanar distance relative to MnFeCoNiCu LDH may be attributed to the larger atomic radius of Au than other metals.

6. What is the catalyst loading? How did the authors calculate the mass activity?

Answer: The catalyst loadings are provided in revised supporting information (**Table S4**), and the mass activity are calculated using the following equation. We have added the catalyst loading and the calculated process of mass activity in the revised supporting information.

Table S4. The loading amounts of Au_{SA}-MnFeCoNiCu LDH and MnFeCoNiCu LDH.

Sample	Au _{SA} -MnFeCoNiCu LDH	MnFeCoNiCu LDH
Loading of catalyst (mg/cm ²)	0.99	1.02

Mass activity calculation:

$$\text{Mass activity} = j/m$$

Here j is the current density at a certain voltage and m is the catalyst loading.

7. From Fig. 3e, why did both materials in the case $R\eta/j$ exhibit the opposite trend? It is not clear, the authors should provide more clear explanation in this section.

Answer: Thanks for your insightful question. In the original manuscript, the parameter

of $R_{\eta/j}$ is utilized to evaluate the mass transfer at high current density, which is cited from the previous work. However, we find that this parameter is not only related to mass transfer, but also affected by iR compensation. As shown in **Figure R6**, the measured $R_{\eta/j}$ value of Au_SA-MnFeCoNiCu LDH with different iR compensation (95%, 80% and 0%) are very different, and even exhibits the opposite trend, suggesting that the parameter of $R_{\eta/j}$ may be not an objective and rigorous descriptor to assess the reaction kinetics and mass transfer. We apologize for misusing the parameter to evaluate the kinetics in the original manuscript. Therefore, to avoid misleading to the readers, we decide to delete the parameter and the related discussion in revision.

Figure R6. Ratios of $\Delta\eta/\Delta\log|j|$ for Au_SA-MnFeCoNiCu LDH with different compensations at different current density ranges.

8. What is the solution resistance? What is the iR corrections percentage?

Answer: The solution resistance is 2.0 Ω , and the iR correction in this work is 95%. We have added the detailed description about the solution resistance and iR correction in the revised manuscript.

Changes in the revised manuscript

Added discussion: Cyclic voltammetry (CV) curves were collected at the scan rates of 100 mV s^{-1} and 20 mV s^{-1} , and the linear sweep voltammetry (LSV) were measured at 2 mV s^{-1} (95% iR corrected). The solution resistance is 2.0 Ω .

9. The AuSA-MnFeCoNiCu-LDH showed considerably lower Tafel slope than

commercial RuO₂ and IrO₂? What is the Tafel slope of IrO₂? Does the lower Tafel slope indicate a change in the reaction mechanism?

Answer: (1) IrO₂ electrocatalyst is well recognized as commercial OER electrocatalyst due to its good activity and stability, so we use it as a control sample. The Tafel slope of IrO₂ in this work is 59.1 mV dec⁻¹, agreeing well with the previous studies (*Nat. Energy*, 2019, 4, 329–338). We have supplemented the related data in the revised manuscript, as shown in **Figure 3c**. (2) The previous studies have shown that the Tafel slope reflects the reaction kinetics, and the lower Tafel slope implies faster reaction kinetics (*Nat. Commun.*, 2022, 13, 2191). The accelerated reaction kinetics may be induced by the change of the reaction mechanism, which has been confirmed by many reported works (*Adv. Funct. Mater.*, 2022, 32(28):211215; *Joule*, 2021, 5(8):2164–2176; *Nat Energy*, 2019, 4, 329–338).

Figure 3c. Tafel plots of Au_{SA}-MnFeCoNiCu LDH, MnFeCoNiCu LDH and IrO₂.

10. A few recent works of literature that are relevant to this work need to be cited in the "Introduction" and "Results" sections, such as "J. Mater. Chem. A, 2021, 9, 16841-16851 and J. Mater. Chem. A, 2020, 8, 11938-11947".

Answer: Thank you for your suggestion. We have carefully read the recommended literatures e, which are very relevant to this work, so we cite these works in the "Introduction" and "Results" sections. Please see **Ref. 23**, **Ref. 43** in the revised manuscript.

11. How about the HER activity of Au_{SA}-MnFeCoNiCu-LDH? I suggested that the authors show it in the supporting information.

Answer: Thank you for your valuable suggestion. We have supplemented the HER performance of Au_{SA}-MnFeCoNiCu LDH and MnFeCoNiCu LDH in the revised supporting information, as shown in **Figure S34**. The result indicates that the HER activity of Au_{SA}-MnFeCoNiCu LDH is not very good, which is possibly attributed to the poor conductivity for LDH materials, agreeing well with the reported works (*Nano-Micro Lett.*, 2020, 12, 86; *Chem. Soc. Rev.*, 2021,50, 8790).

Figure S34. HER performance of Au_{SA}-MnFeCoNiCu LDH. (a) LSV curves, (b) Tafel plots, (c) C_{dl} plots, and (d) EIS curves.

Changes in the revised manuscript

Added discussion: Moreover, the HER performance of Au_{SA}-MnFeCoNiCu LDH and MnFeCoNiCu LDH were also investigated, as illustrated in **Supplementary note 3** and **Figure S34**.

Revisions other than reviewer comments: We found that the original ^{18}O isotope experiment was not rigorous. The $^{16}\text{O}^{18}\text{O}$ product in the original manuscript may not only originate from the lattice oxygen of the electrocatalyst, but also come from the electrolyte that contains ^{18}O from ^{18}O -labeled H_2O and ^{16}O from KOH . Therefore, we redesign a new and rigorous isotope test in the revised manuscript, which is based on the previous studies (*Nat. Commun.*, 2020, 11, 4066; *Angew. Chem. Int. Ed.* 2021, 60, 27126-27134). First, $\text{Au}_{\text{SA}}\text{-MnFeCoNiCu}$ LDH and MnFeCoNiCu LDH were activated by electrochemical CV method in 1.0 M ^{18}O -labeled KOH electrolyte. Then, the ^{18}O -labeled electrocatalysts were carried out OER tests in 1.0 M KOH solution with regular H_2O . Hence, the collected ^{18}O -labeled products such as $^{18}\text{O}^{16}\text{O}$ and $^{18}\text{O}_2$ measured by mass spectrum must derive from the lattice oxygen. The updated results are shown in **Figure 4f** of the revision. Although we redesign the isotope experiment, the new results are consistent with the original conclusions, that is, $\text{Au}_{\text{SA}}\text{-MnFeCoNiCu}$ LDH tends to follow the LOM during OER, while MnFeCoNiCu LDH favors the AEM.

Figure 4f. Mass spectrometric results by ^{18}O isotope-labeling experiments. The signals were normalized by initial intensity of $^{16}\text{O}_2$.

Changes in the revised manuscript

Original discussion: To directly evidence the participation of lattice oxygen in the OER process, the ^{18}O -labeled gas chromatography-mass spectrometer (GC-MS) measurements were performed, and the test details were described in the experimental section. The results shown in **Figure 4f** validate the existence of $^{18}\text{O}_2$, $^{16}\text{O}^{18}\text{O}$ and $^{18}\text{O}_2$

in the final gas production, suggesting that both Au_{SA}-MnFeCoNiCu LDH and MnFeCoNiCu LDH involve in LOM. However, the content of ¹⁶O¹⁸O product in Au_{SA}-MnFeCoNiCu LDH (80.5%) is significantly higher than that in MnFeCoNiCu LDH (12.7%), implying that more lattice oxygen in Au_{SA}-MnFeCoNiCu LDH was involved in the OER reaction compared with that in MnFeCoNiCu LDH.

Revised discussion: To further reveal the participation of lattice oxygen in the OER process, the ¹⁸O-labeled gas chromatography-mass spectrometer (GC-MS) measurements were performed, and the test details were described in the experimental section. Both Au_{SA}-MnFeCoNiCu LDH and MnFeCoNiCu LDH were firstly activated by electrochemical CV method in ¹⁸O-labeled KOH electrolyte, and then carried out OER test in 1.0 M KOH with regular H₂O. The collected oxygen products measured by GC-MS (**Figure 4f**) validate the existence of ¹⁸O-labeled products such as ¹⁶O¹⁸O and ¹⁸O₂ for both samples, suggesting that both Au_{SA}-MnFeCoNiCu LDH and MnFeCoNiCu LDH involve in LOM. However, the content of ¹⁶O¹⁸O product in Au_{SA}-MnFeCoNiCu LDH is significantly higher than that in MnFeCoNiCu LDH, implying that more lattice oxygen in Au_{SA}-MnFeCoNiCu LDH was involved in the OER reaction compared with that in MnFeCoNiCu LDH.

Added discussion in experimental section: ¹⁸O isotope labeling experiments: Both Au_{SA}-MnFeCoNiCu LDH and MnFeCoNiCu LDH were firstly activated by electrochemical CV method (0-0.8 V vs. Hg/HgO, 10 CV cycles) in ¹⁸O-labeled KOH electrolyte, and then carried out OER test (10 mA cm⁻² for 30 min) in 1.0 M KOH with regular H₂O. Subsequently, the gas was collected for gas chromatography-mass spectrometry (GC-MS) analysis.

All the modifications were marked in red color in the revised manuscript.

REVIEWERS' COMMENTS

Reviewer #1 (Remarks to the Author):

The authors have addressed most of my concerns, although there may be other more active OER motifs in the DFT section that have not been studied. Overall, it could be considered for publication.

Reviewer #2 (Remarks to the Author):

I think that the present manuscript is adequate for the publication in this journal.

Reviewer #3 (Remarks to the Author):

Comments to authors

The authors have revised the manuscript carefully and provided all the additional data/discussion in the revised manuscript. So, I hope this manuscript is acceptable with minor corrections as follows.

1. Authors can add the EDS elemental quantification for all samples (Figure S36) in the supporting information.
2. Similarly, authors can add the LSV of AuSA MnFeCoNiCu LDH in Figure S38 to easily compare overpotentials.
3. There are some typos in the manuscript and supporting information text.
4. Authors can follow the same style for all the units (e.g., h \diamond hours).
5. Authors can provide all the details in the figure captions, such as scan rates (for LSV and CV), applied potential and frequency range (for EIS), etc.
6. Authors did not follow the same font style/size for the text in the manuscript and supporting information. I recommend that authors carefully check the whole manuscript and supporting information and correct all the errors.

REVIEWERS' COMMENTS

Reviewer #1 (Remarks to the Author):

The authors have addressed most of my concerns, although there may be other more active OER motifs in the DFT section that have not been studied. Overall, it could be considered for publication

Reviewer #2 (Remarks to the Author):

I think that the present manuscript is adequate for the publication in this journal.

Reviewer #3 (Remarks to the Author):

Comments to authors

The authors have revised the manuscript carefully and provided all the additional data/discussion in the revised manuscript. So, I hope this manuscript is acceptable with minor corrections as follows.

1. Authors can add the EDS elemental quantification for all samples (Figure S36) in the supporting information.

Anwer: Thank you for your suggestion. We have added the EDS quantification information for all samples in Figure S36 in the revised Supporting Information, see Table S6.

Table S6. EDS results of Ru_{SA}-MnFeCoNiCu LDH, Ag_{SA}-MnFeCoNiCu LDH and Pt_{SA}-MnFeCoNiCu LDH.

	Mn (at%)	Fe (at%)	Co (at%)	Ni (at%)	Cu (at%)	Ru (at%)	Ag (at%)	Pt (at%)	O (at%)
Ru _{SA} -	3.23	6.54	6.42	8.71	7.38	0.30	-	-	67.42

MnFeCoNiCu LDH									
Ag _{SA} - MnFeCoNiCu LDH	3.96	7.21	6.55	8.56	7.93	-	0.31	-	65.48
Pt _{SA} - MnFeCoNiCu LDH	3.58	6.99	7.29	8.14	7.67	-	-	0.28	66.05

2. Similarly, authors can add the LSV of Au_{SA}-MnFeCoNiCu LDH in Figure S38 to easily compare overpotentials.

Answer: Thank you for your suggestion. We have added the LSV curves of Au_{SA}-MnFeCoNiCu LDH in Figure S38 for readers to easily compare overpotentials.

3. There are some typos in the manuscript and supporting information text.

Answer : Thanks for your reminding. We have made changes in the revised manuscript and checked the full text.

4. Authors can follow the same style for all the units (e.g., h \diamond hours).

Answer : Thank you for your careful reading. We have checked the entire text to ensure uniformity of unit styles.

5. Authors can provide all the details in the figure captions, such as scan rates (for LSV and CV), applied potential and frequency range (for EIS), etc.

Answer : Thank you for your suggestion. We have now provided detailed electrochemical test information in all the figure captions.

6. Authors did not follow the same font style/size for the text in the manuscript and supporting information. I recommend that authors carefully check the whole manuscript and supporting information and correct all the errors.

Answer : Thank you for your carefully scrutiny. We checked the entire manuscript and supporting information, and the revised version of the manuscript and supporting information all use the same font and font size.